# LOGICAL OPTIONS FRAMEWORK

## ABSTRACT

Learning composable policies for environments with complex rules and tasks is a challenging problem. We introduce a hierarchical reinforcement learning framework called the *Logical Options Framework* (LOF) that learns policies that are *satisfying*, *optimal*, and *composable*. LOF efficiently learns policies that satisfy tasks by representing the task as an automaton and integrating it into learning and planning. We provide and prove conditions under which LOF will learn satisfying, optimal policies. And lastly, we show how LOF's learned policies can be composed to satisfy unseen tasks with only 10-50 retraining steps. We evaluate LOF on four tasks in discrete and continuous domains.

## 1 INTRODUCTION

To operate in the real world, intelligent agents must be able to make long-term plans by reasoning over symbolic abstractions while also maintaining the ability to react to low-level stimuli in their environment (Zhang & Sridharan, 2020). Many environments obey rules that can be represented as logical formulae; e.g., the rules a car follows while driving, or a recipe a chef follows to cook a dish. Traditional motion and path planning techniques struggle to formulate plans over these kinds of long-horizon tasks, but hierarchical approaches such as hierarchical reinforcement learning (HRL) can solve lengthy tasks by planning over both the high-level rules and the low-level environment. However, solving these problems involves trade-offs among multiple desirable properties, which we identify as *satisfaction*, *optimality*, and *composability* (described below). Most of today's algorithms sacrifice at least one of these objectives. For example, Reward Machines from Icarte et al. (2018) is satisfying and optimal, but not composable; the options framework (Sutton et al., 1999) is composable and hierarchically optimal, but cannot satisfy specifications. We introduce a new approach called the *Logical Options Framework*, which builds upon the options framework and aims to combine symbolic reasoning and low-level control to achieve satisfaction, optimality, and composability with as few compromises as possible. Furthermore, we show that our framework is compatible with a large variety of domains and planning algorithms, from discrete domains and value iteration to continuous domains and proximal policy optimization (PPO).

**Satisfaction:** An agent operating in an environment governed by rules must be able to satisfy the specified rules. Satisfaction is a concept from formal logic, in which the input to a logical formula causes the formula to evaluate to `True`. Logical formulae can encapsulate rules and tasks like the ones described in Fig. 1, such as "pick up the groceries" and "do not drive into a lake". In this paper, we state conditions under which our method is guaranteed to learn satisfying policies.

**Optimality:** Optimality requires that the agent maximize its expected cumulative reward for each episode. In general, satisfaction can be achieved by rewarding the agent for satisfying the rules of the environment. In hierarchical planning there are several types of optimality, including hierarchical optimality (optimal with respect to the hierarchy) and optimality (optimal with respect to everything). We prove in this paper that our method is hierarchically optimal and, under certain conditions, optimal.

**Composability:** Our method also has the property of composability – once it has learned to satisfy a task, the learned model can be rearranged to satisfy a large variety of related tasks. More specifically, the rules of an environment can be factored into liveness and safety properties, which we discuss in Sec. 3. The learned model can be adapted to satisfy any appropriate new liveness property. A shortcoming of many RL models is that they are not composable – trained to solve one specific task, they are incapable of handling even small variations in the task structure. However, the real world is

a dynamic and unpredictable place, so the ability to automatically reason over as-yet-unseen tasks and rules is a crucial element of intelligence.

"Go **grocery shopping**, pick up the **kid**, and go **home**, unless your partner **calls** telling you that they will pick up the kid, in which case just go **grocery shopping** and then go **home**. And don't drive into the **lake**."

(a) These natural language instructions can be transformed into an FSA, shown in (b).

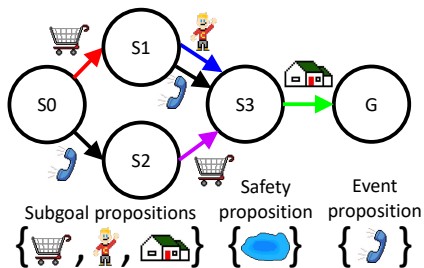

Subgoal propositions      Safety       Event
                       proposition   proposition

(b) The FSA representing the natural language instructions. The propositions are divided into "subgoal", "safety", and "event" propositions.

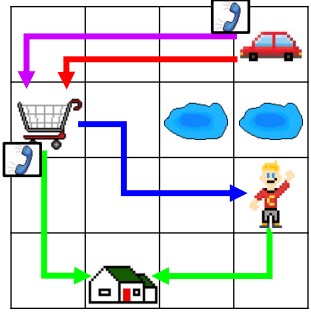

(c) The low-level MDP and corresponding policy that satisfies the instructions.

Figure 1: Many parents face this task after school ends – who picks up the kid, and who gets groceries? The pictorial symbols represent propositions, which are true or false depending on the state of the environment. The arrows in (c) represent subpolicies, and the colors of the arrows match the corresponding transition in the FSA. The boxed phone at the beginning of some of the arrows represents how these subpolicies can occur only after the agent receives a phone call.

The illustrations in Fig. 1 give an overview of our work. The environment is a world with a grocery store, your (hypothetical) kid, your house, and some lakes, and in which you, the agent, are driving a car. The propositions are divided into "subgoals", representing events that can be achieved, such as going grocery shopping; "safety" propositions, representing events that must be avoided (driving into a lake); and "event" propositions, corresponding to events that you have no control over (receiving a phone call) (Fig. 1b). In this environment, you have to follow rules (Fig. 1a). These rules can be converted into a logical formula, and from there into a finite state automaton (FSA) (Fig. 1b). The Logical Options Framework learns an option for each subgoal (illustrated by the arrows in Fig. 1c), and a metapolicy for choosing amongst the options to reach the goal state of the FSA. After learning, the options can be recombined to fulfill other tasks.

## 1.1 CONTRIBUTIONS

We introduce the Logical Options Framework (LOF), which makes four contributions to the hierarchical reinforcement learning literature:

1. The definition of a hierarchical semi-Markov Decision Process (SMDP) that is the product of a logical FSA and a low-level environment MDP.
2. A planning algorithm for learning options and metapolicies for the SMDP that allows for the options to be composed to solve new tasks with only 10-50 retraining steps.
3. Conditions and proofs for achieving satisfaction and optimality.
4. Experiments on a discrete domain and a continuous domain on four tasks demonstrating satisfaction, optimality, and composability.

## 2 BACKGROUND

**Linear Temporal Logic:** We use linear temporal logic (LTL) to formally specify rules (Clarke et al., 2001). LTL formulae are used only indirectly in LOF, as they are converted into automata that the

algorithm uses directly. We chose to use LTL to represent rules because LTL corresponds closely to natural language and has proven to be a more natural way of expressing tasks and rules for engineers than designing FSAs by hand (Kansou, 2019). Formulae $\phi$ have the syntax grammar

$$\phi := p \mid \neg\phi \mid \phi_1 \vee \phi_2 \mid \bigcirc \phi \mid \phi_1 \, \mathcal{U} \, \phi_2$$

where $p$ is a *proposition* (a boolean-valued truth statement that can correspond to objects or events in the world), $\neg$ is negation, $\vee$ is disjunction, $\bigcirc$ is "next", and $\mathcal{U}$ is "until". The derived rules are conjunction ($\wedge$), implication ($\implies$), equivalence ($\leftrightarrow$), "eventually" ($\Diamond\phi \equiv \texttt{True} \, \mathcal{U} \, \phi$) and "always" ($\Box\phi \equiv \neg\Diamond\neg\phi$) (Baier & Katoen, 2008). $\phi_1 \, \mathcal{U} \, \phi_2$ means that $\phi_1$ is true until $\phi_2$ is true, $\Diamond\phi$ means that there is a time where $\phi$ is true and $\Box\phi$ means that $\phi$ is always true.

**The Options Framework:** The options framework is a framework for defining and solving semi-Markov Decision Processes (SMDPs) with a type of macro-action or subpolicy called an option (Sutton et al., 1999). The inclusion of options in an MDP problem turns it into an SMDP problem, because actions are dependent not just on the previous state but also on the identity of the currently active option, which could have been initiated many time steps before the current time.

An option $o$ is a variable-length sequence of actions defined as $o = (\mathcal{I}, \pi, \beta, R_o(s), T_o(s'|s))$. $\mathcal{I} \subseteq \mathcal{S}$ is the initiation set of the option. $\pi : \mathcal{S} \times \mathcal{A} \to [0, 1]$ is the policy the option follows while the option is active. $\beta : \mathcal{S} \to [0, 1]$ is the termination condition. $R_o(s)$ is the reward model of the option. $T_o(s'|s)$ is the transition model. A major challenge in option learning is that, in general, the number of time steps before the option terminates, $k$, is a random variable. With this in mind, $R_o(s)$ is defined as the expected cumulative reward of option $o$ given that the option is initiated in state $s$ at time $t$ and ends after $k$ time steps. Letting $r_t$ be the reward received by the agent at $t$ time steps from the beginning of the option,

$$R_o(s) = \mathbb{E}\big[r_1 + \gamma r_2 + \dots \gamma^{k-1} r_k\big] \tag{1}$$

$T_o(s'|s)$ is the combined probability $p(s', k)$ that option $o$ will terminate at state $s'$ after $k$ time steps:

$$T_o(s'|s) = \sum_{k=1}^{\infty} p(s', k)\gamma^k \tag{2}$$

In the next section, we describe how Eqs. 1 and 2 can be simplified in the context of LOF.

## 3 LOGICAL OPTIONS FRAMEWORK

Here is a brief overview of how we will present our formulation of LOF:

1. The LTL formula is decomposed into liveness and safety properties. The liveness property defines the task specification and the safety property defines the costs for violating rules.

2. The propositions of the formula are divided into three types: subgoals, safety propositions, and event propositions. Subgoals are used to define tasks, and each subgoal is associated with its own option, whose goal is to achieve that subgoal. Safety propositions are used to define rules. Event propositions serve as control flow variables that affect the task.

3. We define an SMDP that is the product of a low-level MDP and a high-level logical FSA.

4. We describe how the logical options can be defined and learned.

5. We present an algorithm for finding the hierarchically optimal policy on the SMDP.

6. We state conditions under which satisfaction of the LTL specification is guaranteed, and we prove that the planning algorithm converges to an optimal policy by showing that the hierarchically optimal SMDP policy is the same as the optimal MDP policy.

**The Logic Formula:** LTL formulae can be translated into Büchi automata using automatic translation tools such as SPOT (Duret-Lutz et al., 2016). All Büchi automata can be decomposed into liveness and safety properties (Alpern & Schneider, 1987). To simplify the formulation, we assume that the LTL formula itself can be divided into liveness and safety formulae, $\phi = \phi_{liveness} \wedge \phi_{safety}$.

For the case where the LTL formula cannot be factored into independent formulae, please see Appendix A. The liveness property describes "things that must happen" to satisfy the LTL formula. It is a task specification, and it is used in planning to determine which subgoals the agent must achieve. The safety property describes "things that can never happen" and is used to define costs for violating the rules. In LOF, the liveness property must be written using a finite-trace subset of LTL called syntactically co-safe LTL (Bhatia et al., 2010), in which the $\square$ ("always") operator is not allowed and $\bigcirc$, $\mathcal{U}$, and $\diamondsuit$ are only used in positive normal form. This way, the liveness property can be satisfied by finite-length sequences of propositions, and the property can be represented as an FSA.

**Propositions:** Propositions are boolean-valued truth statements corresponding to goals, objects, and events in the environment. We distinguish between three types of propositions: subgoals $\mathcal{P}_G$, safety propositions $\mathcal{P}_S$, and event propositions $\mathcal{P}_E$. Subgoal propositions are propositions that must be achieved in order to satisfy the liveness property. They are associated with goals such as "the agent is at the grocery store". They only appear in $\phi_{liveness}$. Each subgoal may only be associated with one state. Note that in general, it may be impossible to avoid having subgoals appear in $\phi_{safety}$. Appendix A describes how to deal with this scenario. Safety propositions are propositions that the agent must avoid – for example, driving into a lake. They only appear in $\phi_{safety}$. Event propositions have a set value that affects the task specification – for example, whether or not a phone call is received. They may occur in $\phi_{liveness}$, and, with some extensions that are described in Appendix A, in $\phi_{safety}$. Although in the fully observable setting, event propositions are somewhat trivial, in the partially observable setting, where the value of the event proposition is revealed to the agent at a random point in time, they are very useful. Our optimality guarantees only apply in the fully observable setting; however, LOF's properties of satisfaction and composability still apply in the partially observable setting. The goal state of the liveness property must be reachable from every other state using only subgoals. This means that no matter what the values of the event propositions are, it is always possible for the agent to satisfy the liveness property. Proposition labeling functions relate states to the set of propositions that are true at that state: $T_{P_G} : \mathcal{S} \to 2^{\mathcal{P}_G}$, $T_{P_S} : \mathcal{S} \to 2^{\mathcal{P}_S}$; for event propositions, a function identifies the set of true propositions, $T_{P_E} : 2^{\mathcal{P}_E} \to \{0, 1\}$.

**Hierarchical SMDP:** LOF works by defining a hierarchical semi-Markov Decision Process (SMDP), learning the options, and then planning over the options. The high-level part of the hierarchy is defined by an FSA specified using LTL. The low level is an environment MDP.

We assume that the high-level LTL specification $\phi$ can be decomposed into a liveness property $\phi_{liveness}$ and a safety property $\phi_{safety}$. The set of propositions $\mathcal{P}$ is the union of the sets of subgoals $\mathcal{P}_G$, safety propositions $\mathcal{P}_S$, and event propositions $\mathcal{P}_E$. We assume that the liveness property can be translated into an FSA $\mathcal{T} = (\mathcal{F}, \mathcal{P}, T_F, R_F, f_0, f_g)$. $\mathcal{F}$ is the set of automaton states; $\mathcal{P}$ is the set of propositions; $T_F$ is the transition function relating the current state and proposition to the next state, $T_F : \mathcal{F} \times \mathcal{P} \times \mathcal{F} \to [0, 1]$. In practice, $T_F$ is deterministic despite our use of probabilistic notation. We assume that there is a single initial state $f_0$ and final state $f_g$, and that the goal state $f_g$ is reachable from every state $f \in \mathcal{F}$ using only subgoals. There is also a reward function that assigns a reward to every state, $R_F : \mathcal{F} \to \mathbb{R}$. In our experiments, we assume that the safety property takes the form $\bigwedge_{p_s \in \mathcal{P}_S} \square \neg p_s$. This simple safety property implies that every safety proposition is not allowed, and that the safety propositions have associated costs, $R_S : 2^{\mathcal{P}_S} \to \mathbb{R}$. $\phi_{safety}$ is not limited to this simple case; the general case is covered in Appendix A.

There is a low-level environment MDP $\mathcal{E} = (\mathcal{S}, \mathcal{A}, R_{\mathcal{E}}, T_E, \gamma)$. $\mathcal{S}$ is the state space and $\mathcal{A}$ is the action space. They can be discrete or continuous. $R_E : \mathcal{S} \times \mathcal{A} \to \mathbb{R}$ is a low-level reward function that characterizes, for example, distance or actuation costs. $R_{\mathcal{E}}$ is a combination of the safety reward function $R_S$ and $R_E$, e.g. $R_{\mathcal{E}}(s, a) = R_E(s, a) + R_S(T_{P_S}(s))$. The transition function of the environment is $T_E : \mathcal{S} \times \mathcal{A} \times \mathcal{S} \to [0, 1]$.

From these parts we define a hierarchical SMDP $\mathcal{M} = (\mathcal{S} \times \mathcal{F}, \mathcal{A}, \mathcal{P}, \mathcal{O}, T_E \times T_P \times T_F, R_{SMDP}, \gamma)$. The hierarchical state space contains two elements: low-level states $\mathcal{S}$ and FSA states $\mathcal{F}$. The action space is $\mathcal{A}$. The set of propositions is $\mathcal{P}$. The set of options (one option associated with each subgoal in $\mathcal{P}_G$) is $\mathcal{O}$. The transition function consists of the low-level environment transitions $T_E$ and the FSA transitions $T_F$. $T_P = T_{P_G} \times T_{P_S} \times T_{P_E}$. We classify $T_P$, relating states to propositions, as a transition function because it helps to determine when FSA transitions occur. The transitions are applied in the order $T_E$, $T_P$, $T_F$. The reward function $R_{SMDP}(f, s, o) = R_F(f)R_o(s)$, so $R_F(f)$ is a weighting on the option rewards. Lastly, the SMDP has the same discount factor $\gamma$ as $\mathcal{E}$. Planning is done on the SMDP in two steps: first, the options $\mathcal{O}$ are learned over $\mathcal{E}$ using an

---

**Algorithm 1** Learning and Planning with Logical Options

---

1: **procedure** LEARNING-AND-PLANNING-WITH-LOGICAL-OPTIONS
2:     Given:
        Propositions $\mathcal{P}$ partitioned into subgoals $\mathcal{P}_G$, safety props $\mathcal{P}_S$, and event props $\mathcal{P}_E$
        Logical FSA $\mathcal{T} = (\mathcal{F}, \mathcal{P}_G \times \mathcal{P}_E, T_F, R_F, f_0, f_g)$ derived from $\phi_{liveness}$
        Low-level MDP $\mathcal{E} = (\mathcal{S}, \mathcal{A}, R_{\mathcal{E}}, T_E, \gamma)$, where $R_{\mathcal{E}}(s,a) = R_E(s,a) + R_S(T_{P_S}(s))$
            combines the environment and safety rewards
        Proposition labeling functions $T_{P_G} : \mathcal{S} \to 2^{\mathcal{P}_G}, T_{P_S} : \mathcal{S} \to 2^{\mathcal{P}_S}$, and
            $T_{P_E} : 2^{\mathcal{P}_E} \to \{0,1\}$
3:     To learn:
4:         Set of options $\mathcal{O}$, one for each subgoal proposition $p \in \mathcal{P}_G$
5:         Metapolicy $\mu(f,s,o)$ along with $Q(f,s,o)$ and $V(f,s)$
6:     **Learn logical options:**
7:     For every $p$ in $\mathcal{P}_G$, learn an option for achieving $p$, $o_p = (\mathcal{I}_{o_p}, \pi_{o_p}, \beta_{o_p}, R_{o_p}, T_{o_p})$
8:         $\mathcal{I}_{o_p} = \mathcal{S}$
9:         $\beta_{o_p} = \begin{cases} 1 & \text{if } p \in T_{P_G}(s) \\ 0 & \text{otherwise} \end{cases}$
10:         $\pi_{o_p}$ = optimal policy on $\mathcal{E}$ with rollouts terminating when $p \in T_{P_G}(s)$
11:         $T_{o_p}(s'|s) = \begin{cases} \mathbb{E}\gamma^k & \text{if } p \in T_{P_G}(s'), \text{where } k \text{ is number of time steps to reach } p \\ 0 & \text{otherwise} \end{cases}$
12:         $R_{o_p}(s) = \mathbb{E}[R_{\mathcal{E}}(s,a_1) + \gamma R_{\mathcal{E}}(s_1,a_2) + \cdots + \gamma^{k-1} R_{\mathcal{E}}(s_{k-1},a_k)]$
13:     **Find a metapolicy $\mu$ over the options:**
14:     Initialize $Q : \mathcal{F} \times \mathcal{S} \times \mathcal{O} \to \mathbb{R}$ and $V : \mathcal{F} \times \mathcal{S} \to \mathbb{R}$ to 0
15:     For $(k,f,s) \in [1,\ldots,n] \times \mathcal{F} \times \mathcal{S}$:
16:         For $o \in \mathcal{O}$:
17:             $Q_k(f,s,o) \leftarrow R_F(f)R_o(s) + \sum_{f' \in \mathcal{F}} \sum_{\bar{p}_e \in 2^{\mathcal{P}_E}} \sum_{s' \in \mathcal{S}} T_F(f'|f, T_{P_G}(s'), \bar{p}_e)$
18:             $T_{P_E}(\bar{p}_e) T_o(s'|s) V_{k-1}(f',s')$
19:         $V_k(f,s) \leftarrow \max_{o \in \mathcal{O}} Q_k(f,s,o)$
20:     $\mu(f,s,o) = \arg\max_{o \in \mathcal{O}} Q(f,s,o)$
21:     **return** Options $\mathcal{O}$, metapolicy $\mu(f,s,o)$ and Q- and value functions $Q(f,s,o), V(f,s)$
22: **end procedure**

---

appropriate policy-learning algorithm such as PPO or Reward Machines. Next, a metapolicy over the task specification $\mathcal{T}$ is found using the learned options and the reward function $R_{SMDP}$.

**Logical Options:** The first step of Alg. 1 is to learn the logical options. We associate every subgoal $p$ with an option $o_p = (\mathcal{I}_{o_p}, \pi_{o_p}, \beta_{o_p}, R_{o_p}, T_{o_p})$. These terms are defined starting at Alg. 1 line 6. Every $o_p$ has a policy $\pi_{o_p}$ whose goal is to reach the state $s_p$ where $p$ is true. Options are learned by training on the environment MDP $\mathcal{E}$ and terminating only when $s_p$ is reached. As we discuss in Sec. 3.1, under certain conditions the optimal option policy is guaranteed to always terminate at the subgoal. This allows us to simplify the transition model of Eq. 2 to the form in Alg. 1 line 11. In the experiments, we further simplify this expression by setting $\gamma = 1$.

**Logical Value Iteration:** After finding the logical options, the next step is to find a policy for FSA $\mathcal{T}$ over the options, as described in Alg. 1 line 13. A value function and Q-function are found for the SMDP using the Bellman update equations:

$$Q_k(f,s,o) \leftarrow R_F(f)R_o(s) + \sum_{f' \in \mathcal{F}} \sum_{\bar{p}_e \in 2^{\mathcal{P}_E}} \sum_{s' \in \mathcal{S}} T_F(f'|f, T_{P_G}(s'), \bar{p}_e)$$

$$T_{P_E}(\bar{p}_e) T_o(s'|s) V_{k-1}(f',s') \tag{3}$$

$$V_k(f,s) \leftarrow \max_{o \in \mathcal{O}} Q_k(f,s,o) \tag{4}$$

Eq. 3 differs from the generic equations for SMDP value iteration in that the transition function has two extra components, $\sum_{f' \in \mathcal{F}} T_F(f'|f, T_P(s'), \bar{p}_e)$ and $\sum_{\bar{p}_e \in 2^{\mathcal{P}_E}} T_{P_E}(\bar{p}_e)$. The equations are

derived from Araki et al. (2019) and the fact that, on every step in the environment, three transitions are applied: the option transition $T_o$, the event proposition "transition" $T_{P_E}$, and the FSA transition $T_F$. Note that $R_o(s)$ and $T_o(s'|s)$ compress the consequences of choosing an option $o$ at a state $s$ from a multi-step trajectory into two real-valued numbers, allowing for more efficient planning.

### 3.1 Conditions for Satisfaction and Optimality

Here we give an overview of the proofs and necessary conditions for satisfaction and optimality. The full proofs and definitions are in Appendix B using the more general formulation of Appendix A.

First, we describe the condition necessary for an optimal option to always reach its subgoal. Let $\pi'(s|s')$ be the optimal goal-conditioned policy for reaching a goal $s'$. If the optimal option policy equals the goal-conditioned policy for reaching the subgoal $s_g$, i.e. $\pi^*(s) = \pi_g(s|s_g)$, then the option will always reach the subgoal. This can also be stated in terms of value functions: let $V^{\pi'}(s|s')$ be the expected return of $\pi'(s|s')$. If $V^{\pi_g}(s|s_g) > V^{\pi'}(s|s') \ \forall s, s' \neq s_g$, then $\pi^*(s) = \pi_g(s|s_g)$. This occurs for example if $-\infty < R_{\mathcal{E}}(s,a) < 0$ and if the episode terminates only when the agent reaches $s_g$. Then $V^{\pi_g}$ is a bounded negative number, and $V^{\pi'}$ for all other states is $-\infty$. We show that if every option is guaranteed to achieve its subgoal, then there must exist at least one sequence of options that satisfies the specification.

We then give the condition for the hierarchically optimal metapolicy $\mu^*(s)$ to always achieve the FSA goal state $f_g$. In our context, hierarchical optimality means that the metapolicy is optimal over the available options. Let $\mu'(f, s|f')$ be the hierarchically optimal goal-conditioned metapolicy for reaching FSA state $f'$. If the hierarchically optimal metapolicy equals the goal-conditioned metapolicy for reaching the FSA goal state $f_g$, i.e. $\mu^*(f, s) = \mu_g(f, s|f_g)$, then $\mu^*(f, s)$ will always reach $f_g$. In terms of value functions: let $V^{\mu'}(f, s|f')$ be the expected return for $\mu'$. If $V^{\mu_g}(f, s|f_g) > V^{\mu'}(f, s|f') \forall f, s, f' \neq f_g$, then $\mu^* = \mu_g$. This occurs if all FSA rewards $R_F(f) > 0$, all environment rewards $-\infty < R_{\mathcal{E}}(s,a) < 0$, and the episode only terminates when the agent reaches $f_g$. Then $V^{\mu_g}$ is a bounded negative number, and $V^{\mu'}$ for all other states is $-\infty$. Because LOF uses the Bellman update equations to learn the metapolicy, the LOF metapolicy will converge to the hierarchically optimal metapolicy.

Consider the SMDP where planning is allowed over low-level actions, and let us call it the "hierarchical MDP" (HMDP) with optimal policy $\pi^*_{HMDP}$. We can then state the final theorem:

**Theorem 3.1.** *Given that the conditions for satisfaction and hierarchical optimality are met, the LOF hierarchically optimal metapolicy $\mu_g$ with optimal option subpolicies $\pi_g$ has the same expected returns as the optimal policy $\pi^*_{HMDP}$ and satisfies the task specification.*

## 4 Experiments & Results

**Experiments:** We performed experiments to demonstrate satisfaction and composability. For the satisfaction experiments, we measure cumulative reward over training steps. Cumulative reward is a proxy for satisfaction, as the environments can only achieve the maximum reward when they satisfy their tasks. For the composability experiments, we take the trained options and record how many metapolicy retraining steps it takes to learn an optimal metapolicy for a new task.

**Environments:** We measure the performance of LOF on two environments. The first environment is a discrete gridworld (Fig. 2a) called the "delivery domain," as it can represent a delivery truck delivering packages to three locations ($a$, $b$, $c$) and having a home base $h$. There are also obstacles $o$ (the black squares). The second environment is called the reacher domain, from OpenAI Gym (Fig. 2d). It is a two-link arm that has continuous state and action spaces. There are four subgoals represented by colored balls: red $r$, green $g$, blue $b$, and yellow $y$. Both environments also have an event proposition called $can$, which represents when the need to fulfill part of a task is cancelled.

**Tasks:** We test satisfaction and composability on four tasks. The first task is a "sequential" task. For the delivery domain, the LTL formula is $\Diamond(a \wedge \Diamond(b \wedge \Diamond(c \wedge \Diamond h))) \wedge \Box \neg o$ – "deliver package $a$, then $b$, then $c$, and then return home $h$. And always avoid obstacles." The next task is the "IF" task (equivalent to the task shown in Fig. 1b): $(\Diamond(c \wedge \Diamond a) \wedge \Box \neg can) \vee (\Diamond a \wedge \Diamond can) \wedge \Box \neg o$ – "deliver package $c$, and then $a$, unless $a$ gets cancelled. And always avoid obstacles". We call the

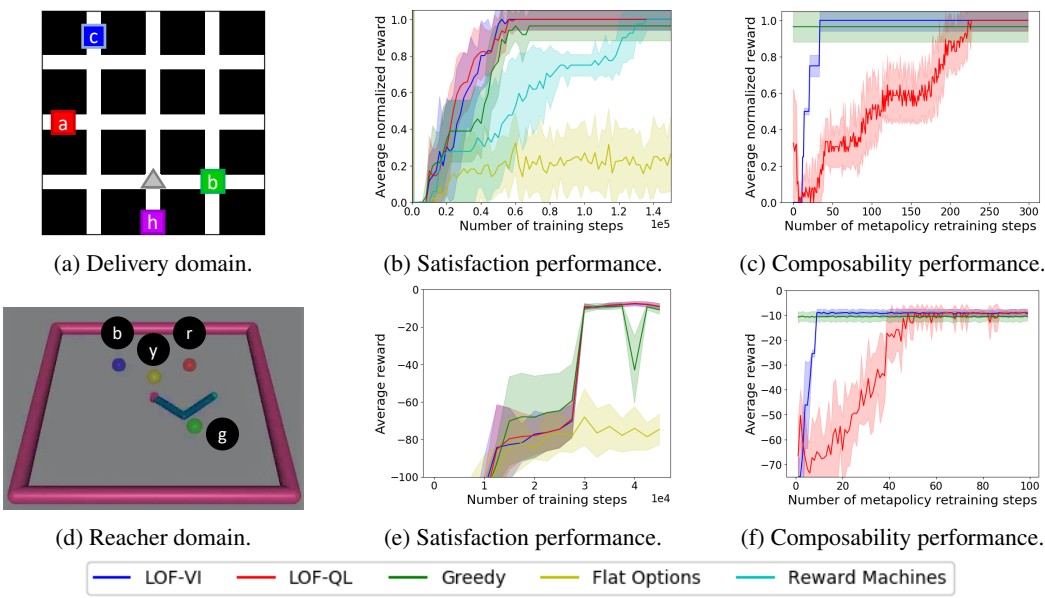

(a) Delivery domain.  (b) Satisfaction performance.  (c) Composability performance.

(d) Reacher domain.  (e) Satisfaction performance.  (f) Composability performance.

LOF-VI  LOF-QL  Greedy  Flat Options  Reward Machines

Figure 2: Performance on the satisfaction and composability experiments, averaged over all tasks. Note that LOF-VI composes new metapolicies in just 10-50 retraining steps. Results for the delivery domain are in the first row, for the reacher domain in the second row. All results, including RM satisfaction performance on the reacher domain, are in Appendix C.6.

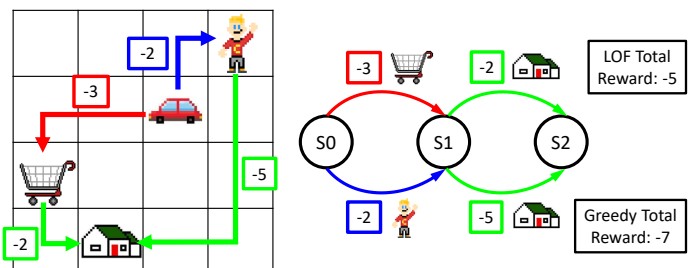

Figure 3: In this environment, the agent must either pick up the kid or go grocery shopping, and then go home. This is equivalent to the OR task. Starting at $S0$, the greedy algorithm picks the next step through the FSA with the lowest cost (in this case, picking up the kid), which leads to a higher overall cost. The LOF algorithm finds the optimal path through the FSA.

third task the "OR" task, $\Diamond((a \vee b) \wedge \Diamond c) \wedge \Box \neg o$ – "deliver package $a$ or $b$, then $c$, and always avoid obstacles". The "composite" task has elements of all three of the previous tasks: $(\Diamond((a \vee b) \wedge \Diamond(c \wedge \Diamond h)) \wedge \Box \neg can) \vee (\Diamond((a \vee b) \wedge \Diamond h) \wedge \Diamond can) \wedge \Box \neg o$. "Deliver package $a$ or $b$, and then $c$, unless $c$ gets cancelled, and then return to home $h$. And always avoid obstacles". The tasks for the reacher environment are equivalent, except that there are no obstacles for the reacher to avoid.

The sequential task is meant to show that planning is efficient and effective even for long-time horizon tasks. The "IF" task shows that the agent's policy can respond to event propositions, such as being alerted that a delivery is cancelled. The "OR" task is meant to demonstrate the optimality of our algorithm versus a greedy algorithm, as discussed in Fig. 3. Lastly, the composite task shows that learning and planning are efficient and effective even for complex tasks.

**Baselines:** We test four baselines against our algorithm. We call our algorithm LOF-VI, short for "Logical Options Framework with Value Iteration," because it uses value iteration for its high-level planning. Our first baseline, LOF-QL, uses Q-learning instead (details can be found in Appendix C.3). Unlike LOF-VI, LOF-QL does not need explicit knowledge of $T_F$, the transition function of the FSA. Greedy is a naive implementation of task satisfaction; it uses its knowledge

of the FSA to select the next subgoal with the lowest cost to attain. This leaves it vulnerable to choosing suboptimal paths through the FSA, as shown in Fig. 3. `Flat Options` uses the regular options framework with no knowledge of the FSA. In other words, its SMDP formulation is not hierarchical – the state space and transition function do not contain high-level states $\mathcal{F}$ or transition function $T_F$. The last baseline is `RM`, short for Reward Machines (Icarte et al., 2018). Whereas LOF learn options to accomplish subgoals, `RM` learns subpolicies for every FSA state. Appendix C.2 discusses the differences between `RM` and LOF in detail.

**Implementation:** For the delivery domain, options were learned using Q-learning with an $\epsilon$-greedy exploration policy. Options were learned simultaneously while switching the option used for exploration at every episode. `RM` was learned using the Q-learning for Reward Machines (QRM) algorithm described in (Icarte et al., 2018). For the reacher domain, options were learned by using proximal policy optimization (PPO) (Schulman et al., 2017) to train goal-oriented policy and value functions, which were represented using a $128 \times 128$ fully connected neural network. Deep-QRM was used to train `RM`. The implementation details are discussed more fully in Appendix C.

### 4.1 RESULTS

**Satisfaction:** Results for the satisfaction experiments, averaged over all four tasks, are shown in Figs. 2b and 2e. (Results on all tasks are in Appendix C.6). As expected, `Flat Options` shows no ability to satisfy tasks, as it has no knowledge of the FSAs. `Greedy` trains as quickly as `LOF-VI` and `LOF-QL`, but its returns plateau before the others because it chooses suboptimal paths in the composite and OR tasks. The difference is small in the reacher domain but still present. `LOF-QL` achieves as high a return as `LOF-VI`, but it is less composable (discussed below). `RM` learns much more slowly than the other methods. This is because for `RM`, a reward is only given for reaching the goal state, whereas in the LOF-based methods, options are rewarded for reaching their subgoals, so during training LOF-based methods have a richer reward function than `RM`. For the reacher domain, `RM` takes an order of magnitude more steps to train, so we left it out of the figure for clarity (see Appendix Fig. 14). However, in the reacher domain, `RM` eventually achieves a higher return than the LOF-based methods. This is because for the reacher domain, we define the subgoals to be spherical regions rather than single states, violating one of the conditions for optimality. Therefore, for example, it is possible that the metapolicy does not take advantage of the dynamics of the arm to swing through the subgoals more efficiently. `RM` does not have this condition and learns a single policy that can take advantage of inter-subgoal dynamics to learn a more optimal policy.

**Composability:** The composability experiments were done on the three composable baselines, `LOF-VI`, `LOF-QL`, and `Greedy`. Appendix C.2 discusses why `RM` is not composable. `Flat Options` is not composable because its formulation does not include the FSA $\mathcal{T}$. Therefore it is completely incapable of recognizing and adjusting to changes in the FSA. The composability results are shown in Figs. 2c and 2f. `Greedy` requires no retraining steps to "learn" a metapolicy on a new FSA – given its current FSA state, it simply chooses the next available FSA state that has the lowest cost to achieve. However, its metapolicy may be arbitrarily suboptimal. `LOF-QL` learns optimal (or in the continuous case, close-to-optimal) policies, but it takes ∼50-250 retraining steps, versus ∼10-50 for `LOF-VI`. Therefore `LOF-VI` strikes a balance between `Greedy` and `LOF-QL`, requiring far fewer steps than `LOF-QL` to retrain, and achieving better performance than `Greedy`.

## 5 RELATED WORK

We distinguish our work from related work in HRL by its possession of three desirable properties – composability, satisfaction, and optimality.

**Not Composable:** The previous work most similar to ours is Icarte et al. (2018; 2019), which introduces a method to solve tasks defined by automata called Reward Machines. Their method learns a subpolicy for every state of the automaton; by transferring rewards between automaton states, they can achieve satisfaction and optimality. However, the learned policies have limited composability because they are specific to the automaton; by contrast, LOF learns a subpolicy for every subgoal, independent of the automaton, and therefore the subpolicies can be arranged to satisfy arbitrary tasks. Another similar work is Logical Value Iteration (LVI) (Araki et al., 2019; 2020). LVI defines a hierarchical MDP and value iteration equations that can find satisfying and optimal

policies; however, the algorithm is limited to discrete domains and has limited composability. A number of HRL algorithms use reward shaping to guide the agent through the states of an automaton (Li et al., 2017; 2019; Camacho et al., 2019; Hasanbeig et al., 2018; Jothimurugan et al., 2019; Shah et al., 2020; Yuan et al., 2019). While these algorithms can guarantee satisfaction and, under certain conditions, optimality, they also have limited composability. Another approach is to use a symbolic planner to find a satisfying sequence of tasks and use an RL agent to learn and execute that sequence of tasks (Gordon et al., 2019; Illanes et al., 2020; Lyu et al., 2019). However, the meta-controllers of Gordon et al. (2019) and Lyu et al. (2019) are not composable as they are trained together with the low-level controllers. Although the work of Illanes et al. (2020) is amenable to transfer learning, it is not composable. Paxton et al. (2017); Mason et al. (2017) use logical constraints to guide exploration, and while these approaches are also satisfying and optimal, they are not composable as the agent is trained for a specific set of rules.

**Not Satisfying:** Most hierarchical frameworks cannot satisfy tasks. Instead, they focus on using state and action abstractions to make learning more efficient (Dietterich, 2000; Dayan & Hinton, 1993; Parr & Russell, 1998; Diuk et al., 2008; Oh et al., 2019). The options framework (Sutton et al., 1999) stands out because of its composability and its guarantee of hierarchical optimality, which is why we based our work off of it. There is also a class of HRL algorithms that builds on the idea of goal-oriented policies that can navigate to nearby subgoals (Eysenbach et al., 2019; Ghosh et al., 2018; Faust et al., 2018). By sampling sequences of subgoals and using a goal-oriented policy to navigate between them, these HRL algorithms can travel much longer distances than a goal-oriented policy can travel on its own. Although these algorithms are "composable" in that they can navigate to arbitrary goals without further training, they are not able to solve tasks, and therefore none of them are capable of satisfying task specifications. Andreas et al. (2017) presents an algorithm for solving simple task "sketches" which is also composable; however, sketches are considerably less expressive than automata and linear temporal logic, which we use.

**Not Optimal:** In HRL, there are at least three types of optimality – hierarchical, recursive, and overall. As defined in Dietterich (2000), the hierarchically optimal policy is the optimal policy given the constraints of the hierarchy, and recursive optimality is when the policy is locally optimal given the policies of its children. For example, the options framework is hierarchically optimal, while MAXQ and abstract MDPs (Gopalan et al., 2017) are recursively optimal. The method described in Kuo et al. (2020) is fully composable, but not optimal as it uses a recurrent neural network to generate a sequence of high-level actions and is therefore not guaranteed to find optimal policies.

# 6 DISCUSSION AND CONCLUSION

In this work we claim that the Logical Options Framework has a unique combination of three properties: satisfaction, optimality, and composability. We state and prove the conditions for satisfaction and optimality in Sec. 3.1. The experimental results confirm our claims while also highlighting some weaknesses. `LOF-VI` achieves optimal or near-optimal policies and trains an order of magnitude faster than the existing work most similar to it, `RM`. However, the optimality condition that each subgoal be associated with exactly one state cannot be met for continuous domains, and therefore `RM` eventually outperforms `LOF-VI`. But even when optimality is not guaranteed, `LOF-VI` is always hierarchically optimal, which is why it outperforms `Greedy` in the composite and OR tasks. Next, the composability experiments show that `LOF-VI` can compose its learned options to accomplish new tasks in very few iterations – about 10-50. Although `Greedy` requires no retraining steps, it is a tiny fraction of the tens of thousands of steps required to learn the original policy. Lastly, we have shown that LOF can learn policies efficiently, and that it can be used with a variety of domains and policy-learning algorithms. In fact, any policy-learning algorithm where it is possible to extract a value (expected cumulative reward) for the policy's execution on the subgoals is fully compatible with LOF, including value iteration and PPO.

Thus, we have proven and demonstrated LOF's features of satisfaction, optimality, and composability, while also reviewing the compromises involved in achieving this goal. We hope that this framework can be useful in practical settings that are governed by complex and changeable rules.

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

# A  FORMULATION OF LOGICAL OPTIONS FRAMEWORK WITH SAFETY AUTOMATON

In this section, we present a more general formulation of LOF than that presented in the paper. In the paper, we make two assumptions that simplify the formulation. The first assumption is that the LTL specification can be divided into two independent formulae, a liveness property and a safety property: $\phi = \phi_{liveness} \wedge \phi_{safety}$. However, not all LTL formulae can be factored in this way. We show how LOF can be applied to LTL formulae that break this assumption. The second assumption is that the safety property takes a simple form that can be represented as a penalty on safety propositions. We show how LOF can be used with arbitrary safety properties.

## A.1  AUTOMATA AND PROPOSITIONS

All LTL formulae can be translated into Büchi automata using automatic translation tools such as SPOT (Duret-Lutz et al., 2016). All Büchi automata can be decomposed into liveness and safety properties (Alpern & Schneider, 1987), so that automaton $\mathcal{W} = \mathcal{W}_{liveness} \times \mathcal{W}_{safety}$. This is a generalization of the assumption that all LTL formulae can be divided into liveness and safety properties $\phi_{liveness}$ and $\phi_{safety}$. The liveness property $\mathcal{W}_{liveness}$ must be an FSA, although this assumption could also be loosened to allow it to be a deterministic Büchi automaton via some minor modifications (allowing multiple goal states to exist and continuing episodes indefinitely, even once a goal state has been reached).

As in the main text, we assume that there are three types of propositions – subgoals $\mathcal{P}_G$, safety propositions $\mathcal{P}_S$, and event propositions $\mathcal{P}_E$. The event propositions have set values and can occur in both $\mathcal{W}_{liveness}$ and $\mathcal{W}_{safety}$. Safety propositions only appear in $\mathcal{W}_{safety}$. Subgoal propositions only appear in $\mathcal{W}_{liveness}$. Each subgoal may only be associated with one state. Note that after writing a specification and decomposing it into $\mathcal{W}_{liveness}$ and $\mathcal{W}_{safety}$, it is possible that some subgoals may unexpectedly appear in $\mathcal{W}_{safety}$. This can be dealt with by creating "safety twins" of each subgoal – safety propositions that are associated with the same low-level states as the subgoals and can therefore substitute for them in $\mathcal{W}_{safety}$.

Subgoals are propositions that the agent must achieve in order to reach the goal state of $\mathcal{W}_{liveness}$. Although event propositions can also define transitions in $\mathcal{W}_{liveness}$, we assume that "achieving" them is not necessary in order to reach the goal state. In other words, we assume that from any state in $\mathcal{W}_{liveness}$, there is a path to the goal state that involves only subgoals. This is because in our formulation, the event propositions are meant to serve as propositions that the agent has no control over, such as receiving a phone call. If satisfaction of the liveness property were to depend on such a proposition, then it would be impossible to guarantee satisfaction. However, if the user is unconcerned with guaranteeing satisfaction, then specifying a liveness property in which satisfaction depends on event propositions is compatible with LOF.

Safety propositions may only occur in $\mathcal{W}_{safety}$ and are associated with things that the agent "must avoid". This is because every state of $\mathcal{W}_{safety}$ is an accepting state (Alpern & Schneider, 1987), so all transitions between the states are non-violating. However, any undefined transition is not allowed and is a violation of the safety property. In our formulation, we assign costs to violations, so that violations are allowed but come at a cost. In practice, it also may be the case that the agent is in a low-level state from which it is impossible to reach the goal state without violating the safety property. In our formulation, satisfaction of the liveness property (but not the safety property) is still guaranteed in this case, as the finite cost associated with violating the rule is less than the infinite cost of not satisfying the liveness property, so the optimal policy for the agent will be to violate the rule in order to satisfy the task (see the proofs, Appendix B). This scenario can be avoided in several ways. For example, do not specify an environment in which it is only possible for the agent to satisfy the task by violating a rule. Or, instead of prioritizing satisfaction of the task, it is possible to instead prioritize satisfaction of the safety property. In this case, satisfaction of the liveness property would not be guaranteed but satisfaction of the safety property would be guaranteed. This could be accomplished by terminating the rollout if a safety violation occurs.

We assume that event propositions are observed – in other words, that we know the values of the event propositions from the start of a rollout. This is because we are planning in a fully observable setting, so we must make this assumption to guarantee convergence to an optimal policy. However, the partially observable case is much more interesting, in which the values of the event propositions are not known until the agent checks or the environment randomly reveals their values. This case is beyond the scope of this paper; however, LOF can still guarantee satisfaction and composability in this setting, just not optimality.

Proposition labeling functions relate states to propositions: $T_{P_G} : \mathcal{S} \rightarrow 2^{\mathcal{P}_G}$, $T_{P_S} : \mathcal{S} \rightarrow 2^{\mathcal{P}_S}$, and $T_{P_E} : 2^{\mathcal{P}_E} \rightarrow \{0, 1\}$.

Given these definitions of propositions, it is possible to define the liveness and safety properties formally. $\mathcal{W}_{liveness} = (\mathcal{F}, \mathcal{P}_G \cup \mathcal{P}_E, T_F, R_F, f_0, f_g)$. $\mathcal{F}$ is the set of states of the liveness property. The propositions can be either subgoals $\mathcal{P}_G$ or event propositions $\mathcal{P}_E$. The transition function relates the current FSA state and active propositions to the next FSA state, $T_F : \mathcal{F} \times 2^{\mathcal{P}_G} \times 2^{\mathcal{P}_E} \times \mathcal{F} \rightarrow [0, 1]$. The reward function assigns a reward to the current FSA state, $R_F : \mathcal{F} \rightarrow \mathbb{R}$. We assume there is one initial state $f_0$ and one goal state $f_g$.

The safety property is a Büchi automaton $\mathcal{W}_{safety} = (\mathcal{F}_S, \mathcal{P}_S \cup \mathcal{P}_E, T_S, R_S, F_0)$. $\mathcal{F}_S$ are the states of the automaton. The propositions can be safety propositions $\mathcal{P}_S$ or event propositions $\mathcal{P}_E$. The transition function $T_S$ relates the current state and active propositions to the next state, $T_S : \mathcal{F}_S \times 2^{\mathcal{P}_S} \times 2^{\mathcal{P}_E} \times \mathcal{F}_S \rightarrow [0, 1]$. The reward function relates the automaton state and safety propositions to rewards (or costs), $R_S : \mathcal{F}_S \times 2^{\mathcal{P}_S} \rightarrow \mathbb{R}$. $F_0$ defines the set of initial states. We do not specify an accepting condition because for safety properties, every state is an accepting state.

## A.2 The Environment MDP

There is a low-level environment MDP $\mathcal{E} = (\mathcal{S}, \mathcal{A}, R_E, T_E, \gamma)$. $\mathcal{S}$ is the state space and $\mathcal{A}$ is the action space. They can be either discrete or continuous. $R_E$ is the low-level reward function that characterizes, for example, time, distance, or actuation costs. $T_E : \mathcal{S} \times \mathcal{A} \times \mathcal{S} \rightarrow [0, 1]$ is the transition function and $\gamma$ is the discount factor. Unlike in the simpler formulation in the paper, we do not combine $R_E$ and the safety automaton reward function $R_S$ in the MDP formulation $\mathcal{E}$.

## A.3 Logical Options

We associate every subgoal $p_g$ with an option $o_{p_g} = (\mathcal{I}_{p_g}, \pi_{p_g}, \beta_{p_g}, R_{p_g}, T_{p_g})$. Every $o_{p_g}$ has a policy $\pi_{p_g}$ whose goal is to reach the state $s_{p_g}$ where $p_g$ is true. Option policies are learned by training on the product of the environment and the safety automaton, $\mathcal{E} \times \mathcal{W}_{safety}$ and terminating training only when $s_{p_g}$ is reached. $R_{\mathcal{E}} : \mathcal{F}_S \times \mathcal{S} \times \mathcal{A} \rightarrow \mathbb{R}$ is the reward function of the product MDP $\mathcal{E} \times \mathcal{W}_{safety}$. There are many reward-shaping policy-learning algorithms that specify how to define $R_{\mathcal{E}}$. In fact, learning a policy for $\mathcal{E} \times \mathcal{W}_{safety}$ is the sort of hierarchical learning problem that many reward-shaping algorithms excel at, including Reward Machines (Icarte et al., 2018) and (Li et al., 2017). This is because in LOF, safety properties are not composable, so using a learning algorithm that is satisfying and optimal but not composable to learn the safety property is appropriate. Alternatively, there are many scenarios where $\mathcal{W}_{safety}$ is a trivial automaton in which each safety proposition is associated with its own state, as we describe in the main paper, so penalties can be assigned to propositions and the state of the agent in $\mathcal{W}_{safety}$ can be ignored.

Note that since the options are trained independently, one limitation of our formulation is that the safety properties cannot depend on the liveness state. In other words, when an agent reaches a new subgoal, the safety property cannot change. However, the workaround for this is not too complicated. First, if the liveness state affects the safety property, this implies that liveness propositions such as subgoals may be in the safety property. In this case, as we described above, the subgoals present in the safety property need to be substituted with "safety twin" propositions. Then during option training, a policy-learning algorithm must be chosen that will learn subpolicies for all of the safety property states, even if those states are only reached after completing a complicated task (for example, all of the subpolicies could be trained in parallel as in (Icarte et al., 2018)). Lastly, during metapolicy learning and during rollouts, when a new option is chosen, the current state of the safety property must be passed to the new option.

---

**Algorithm 2** Learning and Planning with Logical Options

---

1: **procedure** LEARNING-AND-PLANNING-WITH-LOGICAL-OPTIONS
2:     Given:
        Propositions $\mathcal{P}$ partitioned into subgoals $\mathcal{P}_G$, safety propositions $\mathcal{P}_S$, and
            event propositions $\mathcal{P}_E$
        $\mathcal{W}_{liveness} = (\mathcal{F}, \mathcal{P}_G \cup \mathcal{P}_E, T_F, R_F, f_0, f_g)$
        $\mathcal{W}_{safety} = (\mathcal{F}_S, \mathcal{P}_S \cup \mathcal{P}_E, T_S, R_S, F_0)$
        Low-level MDP $\mathcal{E} = (\mathcal{S}, \mathcal{A}, R_E, T_E, \gamma)$
        Proposition labeling functions $T_{P_G} : \mathcal{S} \to 2^{\mathcal{P}_G}$, $T_{P_S} : \mathcal{S} \to 2^{\mathcal{P}_S}$, and
            $T_{P_E} : 2^{\mathcal{P}_E} \to \{0, 1\}$
3:     To learn:
4:         Set of options $\mathcal{O}$, one for each subgoal proposition $p \in \mathcal{P}_G$
5:         Metapolicy $\mu(f, f_s, s, o)$ along with $Q(f, f_s, s, o)$ and $V(f, f_s, s)$
6:     **Learn logical options:**
7:     For every $p$ in $\mathcal{P}_G$, learn an option for achieving $p$, $o_p = (\mathcal{I}_{o_p}, \pi_{o_p}, \beta_{o_p}, R_{o_p}, T_{o_p})$
8:         $\mathcal{I}_{o_p} = \mathcal{S}$
9:         $\beta_{o_p} = \begin{cases} 1 & \text{if } p \in T_{P_G}(s) \\ 0 & \text{otherwise} \end{cases}$
10:        $\pi_{o_p} =$ optimal policy on $\mathcal{E} \times \mathcal{W}_{safety}$ with rollouts terminating when $p \in T_{P_G}(s)$
11:        $T_{o_p}(f'_s, s'|f_s, s) = \begin{cases} \sum_{k=1}^{\infty} p(f'_s, k)\gamma^k & \text{if } p \in T_P(s') \\ 0 & \text{otherwise} \end{cases}$
12:        $R_{o_p}(f_s, s) = \mathbb{E}[\mathcal{R}_{\mathcal{E}}(f_s, s, a_1) + \gamma \mathcal{R}_{\mathcal{E}}(f_{s,1}, s_1, a_2) + \cdots + \gamma^{k-1}\mathcal{R}_{\mathcal{E}}(f_{s,k-1}, s_{k-1}, a_k)]$
13:    **Find a metapolicy $\mu$ over the options:**
14:    Initialize $Q : \mathcal{F} \times \mathcal{F}_S \times \mathcal{S} \times \mathcal{O} \to \mathbb{R}$ and $V : \mathcal{F} \times \mathcal{F}_S \times \mathcal{S} \to \mathbb{R}$ to 0
15:    For $(k, f, f_s, s) \in [1, \dots, n] \times \mathcal{F} \times \mathcal{F}_S \times \mathcal{S}$:
16:        For $o \in \mathcal{O}$:
17:            $Q_k(f, f_s, s, o) \leftarrow R_F(f)R_o(f_s, s) + \sum_{f' \in \mathcal{F}} \sum_{f'_s \in \mathcal{F}_S} \sum_{\bar{p}_e \in 2^{\mathcal{P}_E}} \sum_{s' \in \mathcal{S}}$
18:                $T_F(f'|f, T_{P_G}(s'), \bar{p}_e)T_S(f'_s|f_s, T_{P_S}(s'), \bar{p}_e)\tilde{T}_{P_E}(\bar{p}_e)T_o(s'|s)V_{k-1}(f', f'_s, s')$
19:        $V_k(f, f_s, s) \leftarrow \max_{o \in \mathcal{O}} Q_k(f, f_s, s, o)$
20:    $\mu(f, f_s, s, o) = \arg\max_{o \in \mathcal{O}} Q(f, f_s, s, o)$
21:    **return** Options $\mathcal{O}$, metapolicy $\mu(f, f_s, s, o)$, and $Q(f, f_s, s, o), V(f, f_s, s)$
22: **end procedure**

---

The components of the logical options are defined starting at Alg. 2 line 6. Note that for stochastic low-level transitions, the number of time steps $k$ at which the option terminates is stochastic and characterized by a distribution function. In general this distribution function must be learned, which is a challenging problem. However, there are many approaches to solving this problem; (Abel & Winder, 2019) contains an excellent discussion.

The most notable difference between the general formulation and the formulation in the paper is that the option policy, transition, and reward functions are functions of the safety automaton state $f_s$ as well as the low-level state $s$. This makes Logical Value Iteration more complicated, because in the paper, we could assume we knew the final state of each option (i.e., the state of its associated subgoal $s_g$). But now, although we still assume that the option will terminate at $s_g$, we do not know which safety automaton state it will terminate in, so the transition model must learn a distribution over safety automaton states, and Logical Value Iteration must account for this uncertainty.

### A.4 HIERARCHICAL SMDP

Given a low-level environment $\mathcal{E}$, a liveness property $\mathcal{W}_{liveness}$, a safety property $\mathcal{W}_{safety}$, and logical options $\mathcal{O}$, we can define a hierarchical semi-Markov Decision Process (SMDP) $\mathcal{M} = \mathcal{E} \times \mathcal{W}_{liveness} \times \mathcal{W}_{safety}$ with options $\mathcal{O}$ and reward function $R_{SMDP}$. This SMDP differs significantly from the SMDP in the paper in that the safety property $\mathcal{W}_{safety}$ is now an integral part of the formulation. $R_{SMDP}(f, f_s, s, o) = R_F(f)R_o(f_s, o)$.

### A.5 Logical Value Iteration

A value function and Q-function are found for the SMDP using the Bellman update equations:

$$Q_k(f, f_s, s, o) \leftarrow R_F(f)R_o(f_s, s) + \sum_{f' \in \mathcal{F}} \sum_{f'_s \in \mathcal{F}_S} \sum_{\bar{p}_e \in 2^{\mathcal{P}_E}} \sum_{s' \in \mathcal{S}} T_F(f'|f, T_{P_G}(s'), \bar{p}_e)$$

$$T_S(f'_s|f_s, T_{P_S}(s'), \bar{p}_e)T_{P_E}(\bar{p}_e)T_o(s'|s)V_{k-1}(f', f'_s, s') \tag{5}$$

$$V_k(f, f_s, s) \leftarrow \max_{o \in \mathcal{O}} Q_k(f, f_s, s, o) \tag{6}$$

## B  Proofs and Conditions for Satisfaction and Optimality

The proofs are based on the more general LOF formulation of Appendix A, as results on the more general formulation also apply to the simpler formulation used in the paper.

**Definition B.1.** *Let the reward function of the environment be $R_{\mathcal{E}}(f_s, s, a)$, which is some combination of $R_E(s, a)$ and $R_S(f_s, \bar{p}_s) = R_S(f_s, T_{P_S}(s))$. Let $\pi' : \mathcal{F}_S \times \mathcal{S} \times \mathcal{A} \times \mathcal{S} \to [0, 1]$ be the optimal goal-conditioned policy for reaching a state $s'$. In the case of a goal-conditioned policy, the reward function is $R_{\mathcal{E}}$, and the objective is to maximize the expected reward with the constraint that $s'$ is reached in a finite amount of time. We assume that every state $s'$ is reachable from any state $s$, a standard regularity assumption in MDP literature. Let $V^{\pi'}(f_s, s|s')$ be the optimal expected cumulative reward for reaching $s'$ from $s$ with goal-conditioned policy $\pi'$. Let $s_g$ be the state associated with the subgoal, and let $\pi_g$ be the optimal goal-conditioned policy associated with reaching $s_g$. Let $\pi^*$ be the optimal policy for the environment $\mathcal{E}$.*

**Condition B.1.** *The optimal policy for the option must be the same as the goal-conditioned policy that has subgoal $s_g$ as its goal: $\pi^*(f_s, s) = \pi_g(f_s, s|s_g)$. In other words, $V^{\pi_g}(f_s, s|s_g) > V^{\pi'}(f_s, s|s') \ \forall f_s, s, s' \neq s_g$.*

This condition guarantees that the optimal option policy will always reach the subgoal $s_g$. It can be achieved by setting all rewards $-\infty < R_{\mathcal{E}}(f_s, s, a) < 0$ and terminating the episode only when the agent reaches $s_g$. Therefore the expected return for reaching $s_g$ is a bounded negative number, and the expected return for all other states is $-\infty$.

**Lemma B.2.** *Given that the goal state of $\mathcal{W}_{liveness}$ is reachable from any other state using only subgoals and that there is an option for every subgoal and that all the options meet Condition B.1, there exists a metapolicy that can reach the FSA goal state from any non-trap state in the FSA.*

*Proof.* This follows from the fact that transitions in $\mathcal{W}_{liveness}$ are determined by achieving subgoals, and it is given that there exists an option for achieving every subgoal. Therefore, it is possible for the agent to execute any sequence of subgoals, and at least one of those sequences must satisfy the task specification since the FSA representing the task specification is finite and satisfiable, and the goal state $f_g$ is reachable from every FSA state $f \in \mathcal{F}$ using only subgoals. $\square$

**Definition B.2.** *From Dietterich (2000): A **hierarchically optimal** policy for an MDP or SMDP is a policy that achieves the highest cumulative reward among all policies consistent with the given hierarchy.*

In our case, this means that the hierarchically optimal metapolicy is optimal over the available options.

**Definition B.3.** *Let the expected cumulative reward function of an option $o$ started at state $(f_s, s)$ be $R_o(f_s, s)$. Let the reward function on the SMDP be $R_{SMDP}(f, f_s, s, o) = R_F(f)R_o(f_s, s)$ with $R_F(f) \geq 0^1$. Let $\mu' : \mathcal{F} \times \mathcal{F}_S \times \mathcal{S} \times \mathcal{O} \times \mathcal{F} \to [0, 1]$ be the hierarchically optimal goal-conditioned metapolicy for achieving liveness state $f'$. The objective of the metapolicy is to maximize the reward function $R_{SMDP}$ with the constraint that it reaches $f'$ in a finite number of time*

---

[1]The assumption that $R_{SMDP}(f, f_s, s, o) = R_F(f)R_o(f_s, s)$ and $R_{HMDP}(f, f_s, s, a) = R_F(f)R_{\mathcal{E}}(f_s, s, a)$ can be relaxed so that $R_{SMDP}$ and $R_{HMDP}$ are functions that are monotonic increasing in the low-level rewards $R_o$ and $R_{\mathcal{E}}$, respectively.

*steps.* Let $V^{\mu'}(f, f_s, s|f')$ *be the hierarchically optimal return for reaching* $f'$ *from* $(f, f_s, s)$ *with goal-conditioned metapolicy* $mu'$. *Let* $\mu^*$ *be the hierarchically optimal policy for the SMDP. Let* $f_g$ *be the goal state, and* $\mu_g$ *be the hierarchically optimal goal-conditioned metapolicy for achieving the goal state.*

**Condition B.3.** *The hierarchically optimal metapolicy must be the same as the goal-conditioned metapolicy that has the FSA goal state* $f_g$ *as its goal:* $\mu^*(f, f_s, s) = \mu_g(f, f_s, s|f_g)$. *In other words,* $V^{\mu_g}(f, f_s, s|f_g) > V^{\mu'}(f, f_s, s|f') \;\; \forall f, f_s, s, f' \neq f_g$.

This condition guarantees that the hierarchically optimal metapolicy will always go to the FSA goal state $f_g$ (thereby satisfying the specification). Here is an example of how this condition can be achieved: If $-\infty < R_{\mathcal{E}}(f_s, s, a) < 0 \;\; \forall s$, then $R_o(f_s, s) < 0 \;\; \forall f_s, o, s$. Then if $R_F(f) > 0$ (in our experiments, we set $R_F(f) = 1 \;\; \forall f$), $R_{SMDP}(f, f_s, s, o) = R_F(f)R_o(f_s, s) < 0$, and if the episode only terminates when the agent reaches the goal state, then the expected return for reaching $f_g$ is a bounded negative number, and the expected return for all other states is $-\infty$.

**Lemma B.4.** *From (Sutton et al., 1999): Value iteration on an SMDP converges to the hierarchically optimal policy.*

Therefore, the metapolicy found using the Logical Options Framework converges to a hierarchically optimal metapolicy that satisfies the task specification as long as Conditions B.1 and B.3 are met.

**Definition B.4.** *Consider the SMDP where planning is allowed over the low-level actions instead of the options. We will call this the hierarchical MDP (HMDP), as this MDP is the product of the low-level environment* $\mathcal{E}$, *the liveness property* $\mathcal{W}_{liveness}$, *and the safety property* $\mathcal{W}_{safety}$. *Let* $R_F(f) > 0 \;\forall f$, *and let* $R_{HMDP}(f, f_s, s, a) = R_F(f)R_{\mathcal{E}}(f_s, s, a)$, *and let* $\pi^*_{HMDP}$ *be the optimal policy for the HMDP.*

**Theorem B.5.** *Given Conditions B.1 and B.3, the hierarchically optimal metapolicy* $\mu_g$ *with optimal option policies* $\pi_g$ *has the same expected returns as the HMDP optimal policy* $\pi^*$ *and satisfies the task specification.*

*Proof.* By Condition B.1, every subgoal has an option associated with it whose optimal policy is to go to the subgoal. By Condition B.3, the hierarchically optimal metapolicy will reach the FSA goal state $f_g$. The metapolicy can only accomplish this by going to the subgoals in a sequence that satisfies the task specification. It does this by executing a sequence of options that correspond to a satisfying sequence of subgoals and are optimal in expectation. Therefore, since $R_F(f) > 0 \;\forall f$ and $R_{SMDP}(f, f_s, s, o) = R_F(f)R_o(f_s, s)$, and since the event propositions that affect the order of subgoals necessary to satisfy the task are independent random variables, the expected cumulative reward is a positive linear combination of the expected option rewards, and since all option rewards are optimal with respect to the environment and the metapolicy is optimal over the options, our algorithm attains the optimal expected cumulative reward. $\square$

## C  Experimental Implementation

We discuss the implementation details of the experiments in this section. Because the delivery and reacher domains are analogous, we discuss the delivery domain first in every section and then briefly relate how the same formulation applies to the reacher domain as well. In this section, we use the simpler formulation of the main paper and not the more general formulation discussed in Appendix A.

### C.1  Propositions

The delivery domain has 7 propositions plus 4 composite propositions. The subgoal propositions are $\mathcal{P}_G = \{a, b, c, h\}$. Each of these propositions is associated with a single state in the environment (see Fig. 12a). The safety propositions are $\mathcal{P}_S = \{o, e\}$. $o$ is the obstacle proposition. It is associated with many states – the black squares in Fig. 12a. $e$ is the empty proposition, associated with all of the white squares in the domain. This is the default proposition for when there are no other active propositions. The event proposition is $\mathcal{P}_E = \{can\}$. $can$ is the "cancelled" proposition, representing when one of the subgoals has been cancelled.

To simplify the FSAs and the implementation, we make an assumption that multiple propositions cannot be true at the same state. However, it is reasonable for $can$ to be true at the subgoals, and therefore we introduce 4 composite propositions, $ca = a \wedge can$, $cb = b \wedge can$, $cc = c \wedge can$, $ch = h \wedge can$. These can be counted as event propositions without affecting the operation of the algorithm.

The reacher domain has analogous propositions. The subgoals are $r, g, b, y$ and correspond to $a, b, c, h$. The environment does not contain obstacles $o$ but does have safety proposition $e$, and it also has the event proposition $can$ and the composite propositions $cr, cg, cb, cy$ for when $can$ is true at the same time that a subgoal proposition is true. Another difference is that the subgoal propositions are associated with a small spherical region instead of a single state as in the delivery domain; this is a necessity for continuous domains and unfortunately breaks one of our conditions for optimality because the subgoals are now associated with multiple states instead of a single state. However, the LOF metapolicy will still converge to a hierarchically optimal policy.

## C.2 REWARD FUNCTIONS

Next, we define the reward functions of the physical environment $R_E$, safety propositions $R_S$, and FSA states $R_F$. We realize that often in reinforcement learning, the algorithm designer has no control over the reward functions of the environment. However, in our case, there are no publicly available environments such as OpenAI Gym or the DeepMind Control Suite that we know of that have a high-level FSA built-in. Therefore, anyone implementing our algorithm will likely have to implement their own high-level FSA and define the rewards associated with it.

Our low-level environment reward function $R_E : \mathcal{S} \times \mathcal{A} \rightarrow \mathbb{R}$ is defined to be $-1 \ \forall s, a$. In other words, it is a time/distance cost.

We assign costs to the safety propositions by defining the reward function $R_S : \mathcal{P}_S \rightarrow \mathbb{R}$. All of the costs are 0 except for the obstacle cost, $R_S(o) = -1000$. Therefore, there is a very high penalty for encountering an obstacle.

We define the environment reward function $R_\mathcal{E} : \mathcal{S} \times \mathcal{A} \rightarrow \mathbb{R}$ to be $R_\mathcal{E}(s, a) = R_E(s, a) + R_S(T_P(s))$. In other words, it is the sum of $R_E$ and $R_S$. This reward function meets Condition B.1 for the optimal option policies to always converge to their subgoals.

Lastly, we define $R_F : \mathcal{F} \rightarrow \mathbb{R}$ to be $R_F(f) = 1 \ \forall f$. Therefore the SMDP cost $R_S MDP(f, s, o) = R_o(s)$ and meets Condition B.3 so that the LOF metapolicy converges to the optimal policy.

The reacher environment has analogous reward functions. The safety reward function $R_S(p) = 0 \ \forall p \in \mathcal{P}_S$ because there is no obstacle proposition. Also, the physical environment reward function differs during option training and metapolicy learning. For metapolicy learning, the reward function is $R_E(s, a) = -a^\top a - 0.1$ – a time cost and an actuation cost. During option training, we speed learning by adding the distance to the goal state as a cost, instead of a time cost: $R_E(s, a) = -a^\top a - ||s - s_g||^2$. Although the reward functions and value functions are different, the costs are analogous and lead to good performance as seen in the results. Note that this method can't be used for Reward Machines, because it trains subpolicies for FSA states, and the subgoals for FSA states are not known ahead of time, so distance to subgoal cannot be calculated.

## C.3 ALGORITHM FOR `LOF-QL`

The `LOF-QL` baseline uses Q-learning to learn the metapolicy instead of value iteration. We therefore use "Logical Q-Learning" equations in place of the Logical Value Iteration equations described in Eqs. 3 and 4 in the main text. The algorithm is described in Alg. 3. A benefit of using Q-learning instead of value iteration is that the transition function $T_F$ of the FSA $\mathcal{T}$ does not have to be explicitly known, as the algorithm samples from the transitions rather than using $T_F$ explicitly in the formula. However, as described in the main text, this comes at the expense of reduced composability, as `LOF-QL` takes around $5x$ more iterations to converge to a new metapolicy than `LOF-VI`

---

**Algorithm 3** LOF with $\epsilon$-greedy Q-learning

---

1: **procedure** LOF-Q-LEARNING
2:     Given:
        Propositions $\mathcal{P}$ partitioned into subgoals $\mathcal{P}_G$, safety propositions $\mathcal{P}_S$, and
           event propositions $\mathcal{P}_E$
        Environment MDP $\mathcal{E} = (\mathcal{S}, \mathcal{A}, T_E, R_{\mathcal{E}}, \gamma)$
        Logical options $\mathcal{O}$ with reward models $R_o(s)$ and transition models $T_o(s'|s)$
        Liveness property $\mathcal{T} = (\mathcal{F}, \mathcal{P}_G \cup \mathcal{P}_E, T_F, R_F, f_0, f_g)$ ($T_F$ does not have to be
           explicitly known if it can be sampled from a simulator)
        Learning rate $\alpha$, exploration probability $\epsilon$
        Number of training episodes $n$, episode length $m$
3:     To learn:
4:         Metapolicy $\mu(f, s, o)$ along with $Q(f, s, o)$ and $V(f, s)$
5:     **Find a metapolicy $\mu$ over the options:**
6:     Initialize $Q : \mathcal{F} \times \mathcal{S} \times \mathcal{O} \to \mathbb{R}$ and $V : \mathcal{F} \times \mathcal{S} \to \mathbb{R}$ to 0
7:     For $k \in [1, \ldots, n]$:
8:         Initialize FSA state $f \leftarrow 0$, $s$ a random initial state from $\mathcal{E}$
9:         $\bar{p}_e \sim T_{P_E}()$
10:       For $j \in [1, \ldots, m]$:
11:         With probability $\epsilon$ let $o$ be a random option; otherwise, $o \leftarrow \underset{o' \in \mathcal{O}}{\arg\max}\, Q(f, s, o')$
12:         $s' \sim T_o(s)$
13:         $f' \sim T_F(T_{P_G}(s'), \bar{p}_e, f)$
14:         $Q_k(f, s, o) \leftarrow Q_{k-1}(f, s, o) + \alpha\big(R_F(f)R_o(s) + \gamma V(f', s') - Q_{k-1}(f, s, o)\big)$
15:         $V_k(f, s) \leftarrow \underset{o' \in \mathcal{O}}{\max}\, Q_k(f, s, o')$
16:         $f \leftarrow f'$
17:     $\mu(f, s, o) = \underset{o \in \mathcal{O}}{\arg\max}\, Q(f, s, o)$
18:     **return** Options $\mathcal{O}$, metapolicy $\mu(f, s, o)$ and Q- and value functions $Q(f, s, o), V(f, s)$
19: **end procedure**

---

does. Let $Q_0(f, s, o)$ be initialized to be all 0s. The Q update formulas are given in Alg. 3 lines 14 and 15.

## C.4   COMPARISON OF LOF AND REWARD MACHINES

Figs. 4, 5, 6, and 7 give a visual overview of how LOF and Reward Machines work, and hopefully illustrate how they differ.

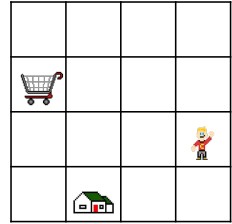

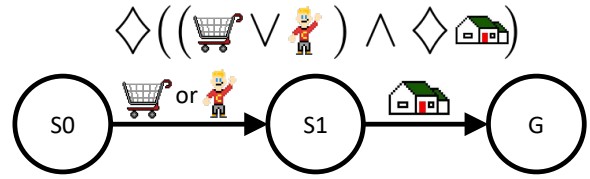

Go grocery shopping OR pick up the kid, then go home.

(a) Environment MDP $\mathcal{E}$.

(b) Liveness property $\mathcal{T}$. The natural language rule can be represented as an LTL formula which can be translated into an FSA.

Figure 4: LOF and RM both require an environment MDP $\mathcal{E}$ and an automaton $\mathcal{T}$ that specifies a task.

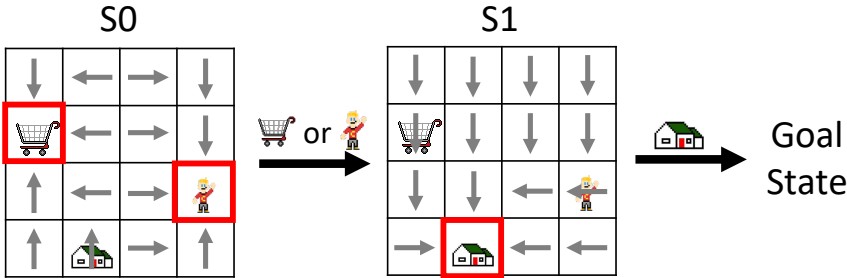

Figure 5: In RM, subpolicies are learned for each state of the automaton. In this case, in state $S0$, a subpolicy is learned that goes either to the shopping cart of the kid, whichever is closer. In state $S1$, the subpolicy goes to the house.

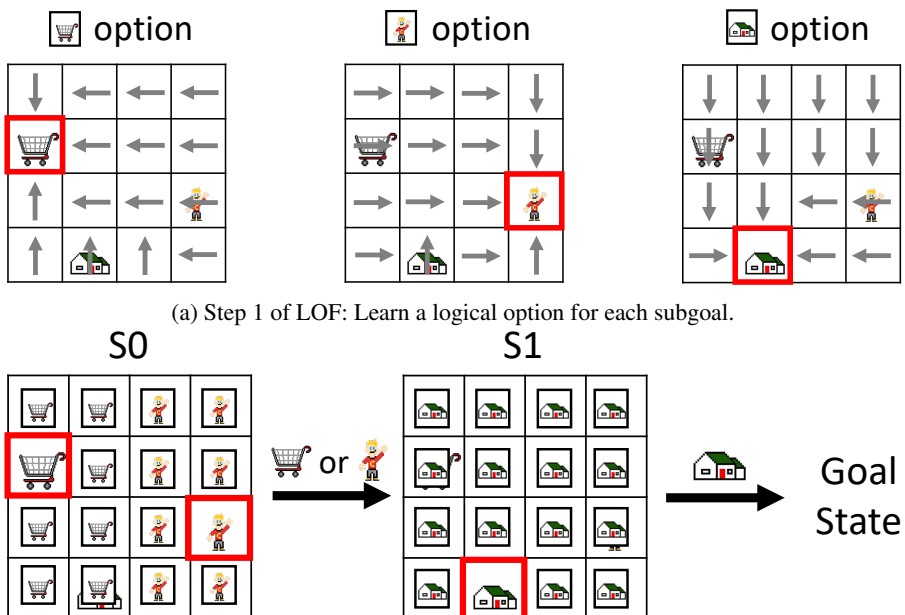

(a) Step 1 of LOF: Learn a logical option for each subgoal.

(b) Step 2 of LOF: Use Logical Value Iteration to find a metapolicy that satisfies the liveness property. In this image, the boxed subgoals indicate that the corresponding option is the optimal option to take from that low-level state. The policy ends up being the same as RM's policy – in state $S0$, the optimal metapolicy chooses the "grocery shoppping" option if the grocery cart is closer and the "pick up kid" option if the kid is closer. In the state $S1$, the optimal metapolicy is to always choose the "home" option.

Figure 6: LOF has two steps. In (a) the first step, logical options are learned for each subgoal. In (b) the second step, a metapolicy is found using Logical Value Iteration.

## C.5 TASKS

We test the environments on four tasks, a "sequential" task (Fig. 8), an "IF" task (Fig. 9), an "OR" task (Fig. 10), and a "composite" task (Fig. 11). The reacher domain has the same tasks, expect $r, g, b, y$ replace $a, b, c, h$, and there are no obstacles $o$. Note that in the LTL formulae, $\Box!o$ is the safety property $\phi_{safety}$; the preceding part of the formula is the liveness property $\phi_{liveness}$ used to construct the FSA.

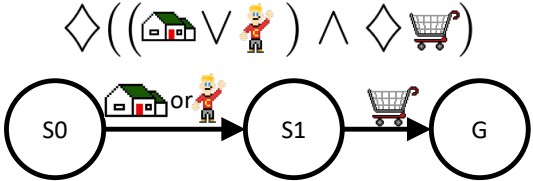

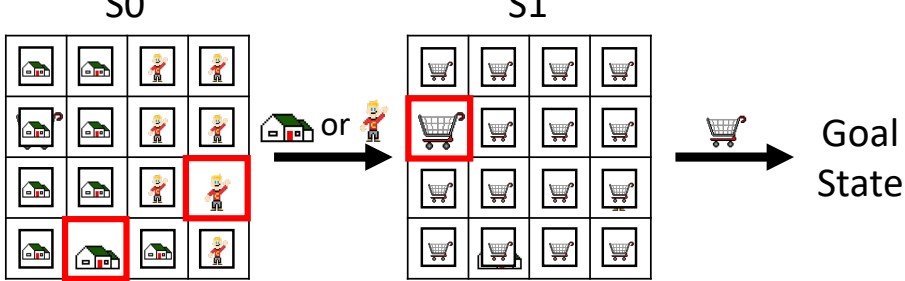

(a) LOF can easily solve this new liveness property without training new options.

(b) Logical Value Iteration can be used to find a metapolicy on the new task without the need to retrain the logical options. A new metapolicy can be found in 10-50 iterations. The new policy finds that in state $S0$, "home" option is optimal if the agent is closer to "home", and the "kid" option is optimal if the agent is closer to "kid". In state $S1$, the "grocery shopping" option is optimal everywhere.

Figure 7: What distinguishes LOF from `RM` is that the logical options of LOF can be easily composed to solve new tasks. In this example, the new task is to go home or pick up the kid, then go grocery shopping. Logical Value Iteration can find a new metapolicy in 10-50 iterations without needing to relearn the options.

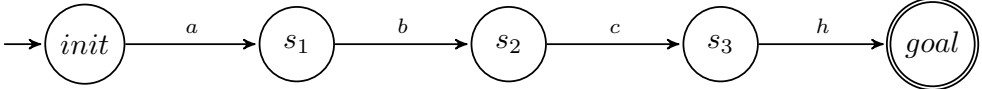

Figure 8: FSA for the sequential task. The LTL formula is $\Diamond(a \wedge \Diamond(b \wedge \Diamond(c \wedge \Diamond h))) \wedge \Box!o$. The natural language interpretation is "Deliver package $a$, then $b$, then $c$, and then return home $h$. And always avoid obstacles $o$".

## C.6 FULL EXPERIMENTAL RESULTS

For the satisfaction experiments for the delivery domain, 10 policies were trained for each task and for each baseline. Training was done for 1600 episodes, with 100 steps per episode. Every 2000 training steps, the policies were tested on the domain and the returns recorded. For this discrete domain, we know the minimum and maximum possible returns for each task, and we normalized the returns using these minimum and maximum returns. The error bars are the standard deviation of the returns over the 10 policies' rollouts.

For the satisfaction experiments for the reacher domain, a single policy was trained for each task and for each baseline. The baselines were trained for 900 epochs, with 50 steps per epoch. Every 2500 training steps, each policy was tested by doing 10 rollouts and recording the returns. For the `RM` baseline, training was for 1000 epochs with 800 steps per epoch, and the policy was tested every 8000 training steps. Because we don't know the minimum and maximum rewards for each task, we did not normalize the returns. The error bars are the standard deviation over the 10 rollouts for each baseline.

For the composability experiments, a set of options was trained once, and then metapolicing training using `LOF-VI`, `LOF-QL`, and `Greedy` was done for each task. Returns were recorded at every

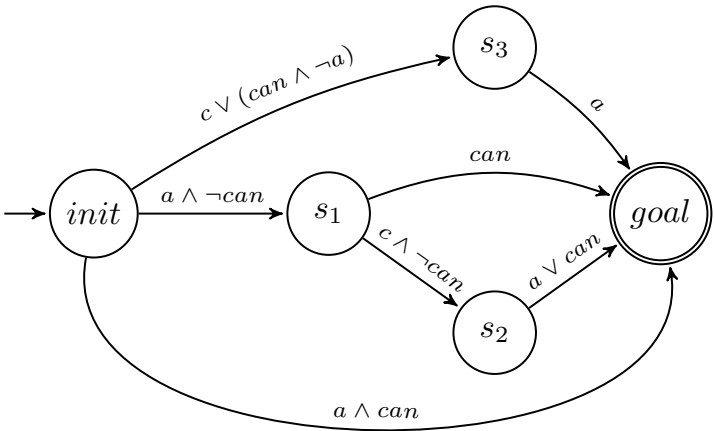

Figure 9: FSA for the IF task. The LTL formula is $(\Diamond(c \wedge \Diamond a) \wedge \Box!can) \vee (\Diamond a \wedge \Diamond can) \wedge \Box!o$. The natural language interpretation is "Deliver package $c$, and then $a$, unless $a$ gets cancelled. And always avoid obstacles $o$".

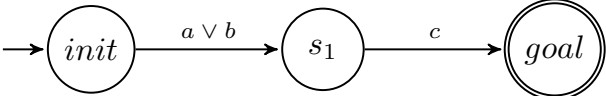

Figure 10: FSA for the OR task. The LTL formula is $\Diamond((a \vee b) \wedge \Diamond c) \wedge \Box!o$. The natural language interpretation is "Deliver package $a$ or $b$, then $c$, and always avoid obstacles $o$".

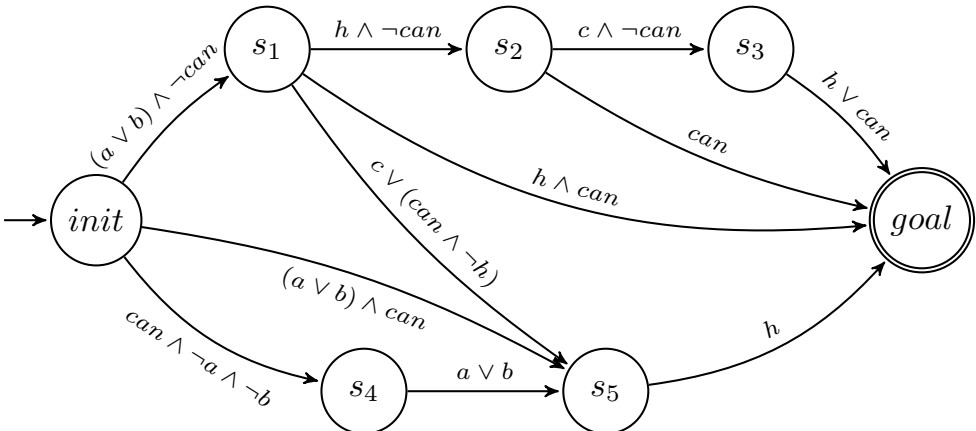

Figure 11: FSA for the composite task. The LTL formula is $(\Diamond((a \vee b) \wedge \Diamond(c \wedge \Diamond h)) \wedge \Box!can) \vee (\Diamond((a \vee b) \wedge \Diamond h) \wedge \Diamond can) \wedge \Box!o$. The natural language interpretation is "Deliver package $a$ or $b$, and then $c$, unless $c$ gets cancelled, and then return to home $h$. And always avoid obstacles".

training step by rolling out each baseline 10 times. The error bars are the standard deviations on the 10 rollouts.

Code and videos of the domains and tasks are in the supplement.

# D FURTHER DISCUSSION

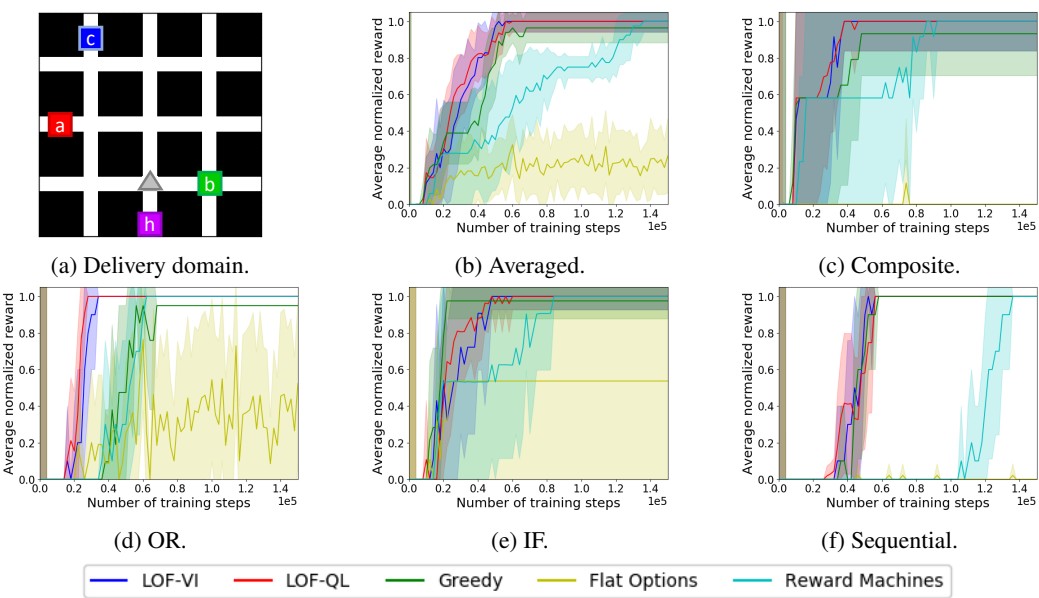

Figure 12: All satisfaction experiments on the delivery domain. Notice how for the composite and OR tasks (Figs. 12c and 12d), the `Greedy` baseline plateaus before `LOF-VI` and `LOF-QL`. This is because `Greedy` chooses a suboptimal path through the FSA, whereas `LOF-VI` and `LOF-QL` find an optimal path. Also, notice that `RM` takes many more training steps to achieve the optimal cumulative reward. This is because for `RM`, the only reward signal is from reaching the goal state. It takes a long time for the agent to learn an optimal policy from such a sparse reward signal. This is particularly evident for the sequential task (Fig. 12f), which requires the agent to take a longer sequence of actions/FSA states before reaching the goal. The options-based algorithms train much faster because when training the options, the agent receives a reward for reaching each subgoal, and therefore the reward signal is much richer.

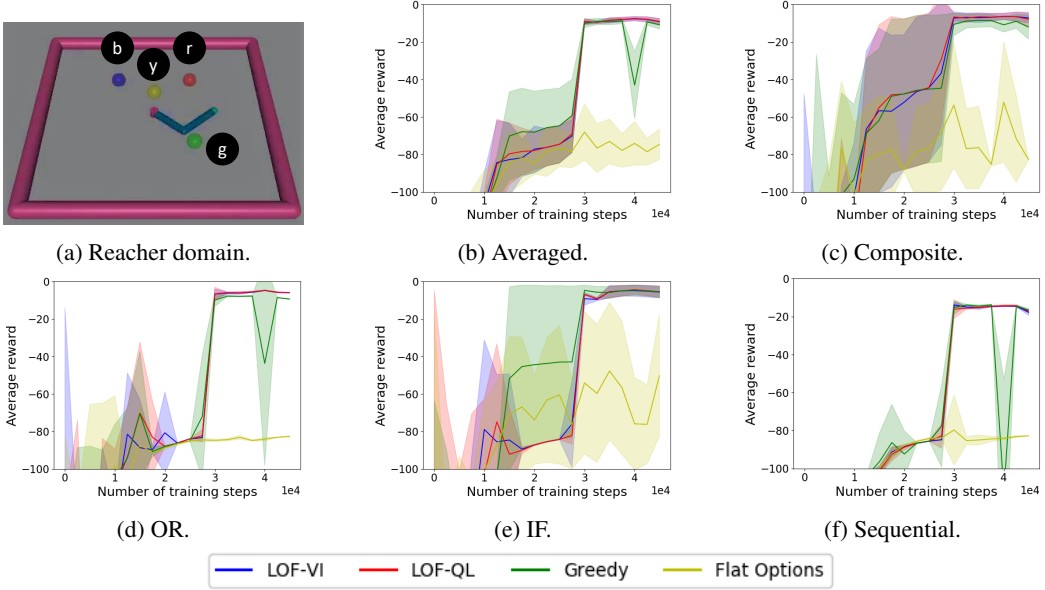

Figure 13: Satisfaction experiments for the reacher domain, without `RM` results. The results are equivalent to the results on the delivery domain.

**What happens when incorrect rules are used?** One benefit of representing the rules of the environment as LTL formulae/automata is that these forms of representing rules are much more in-

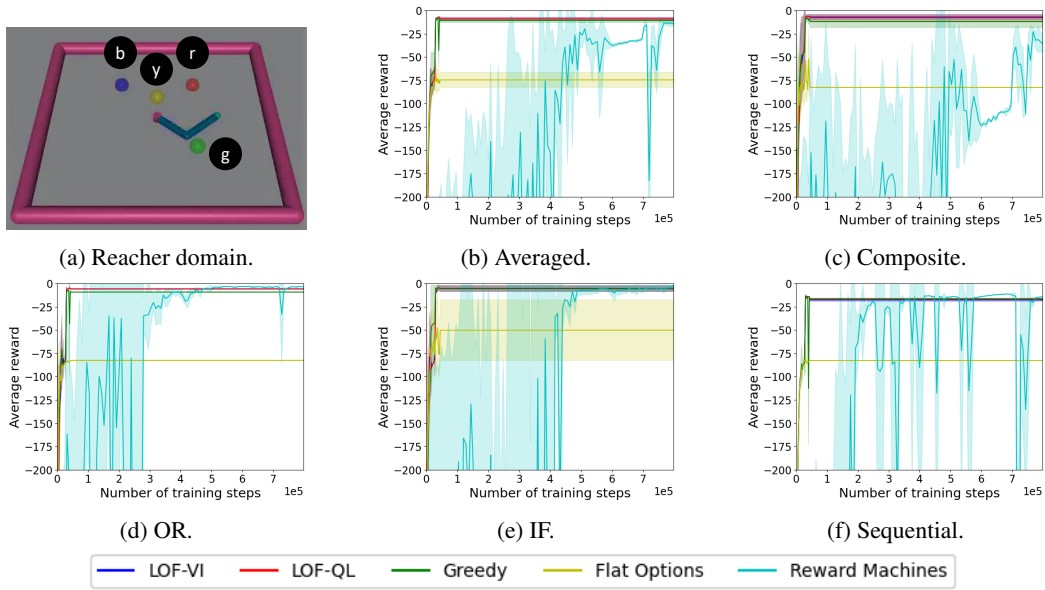

Figure 14: Satisfaction experiments for the reach domain, including RM results. RM takes significantly more training steps to train than the other baselines, although it eventually reaches and surpasses the cumulative reward of the other baselines. This is because for the continuous domain, we violate some of the conditions required for optimality when using the Logical Options Framework – in particular, the condition that each subgoal is associated with a single state. In a continuous environment, this condition is impossible to meet, and therefore we made the subgoals small spherical regions, and we only made the subgoals associated with specific Cartesian coordinates and not velocities (which are also in the state space). Meanwhile, the optimality conditions of RM are looser and were not violated, which is why it achieves a higher final cumulative reward.

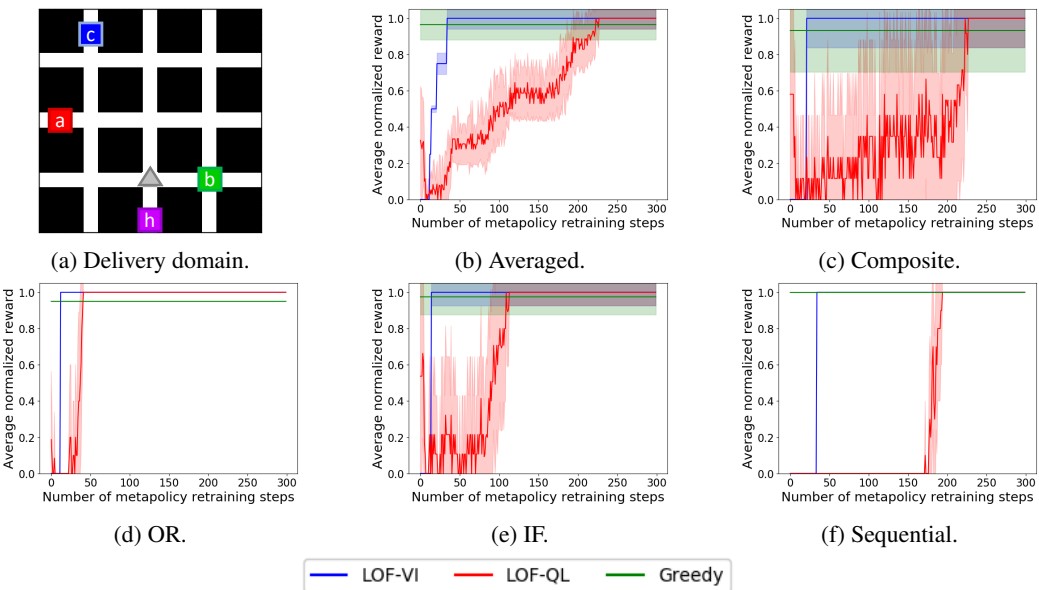

Figure 15: All composability experiments for the delivery domain.

terpretable than alternatives (such as neural nets). Therefore, if an agent's learned policy has bad behavior, a user of LOF can inspect the rules to see if the bad behavior is a consequence of a bad rule specification. Furthermore, one of the consequences of composability is that any modifications

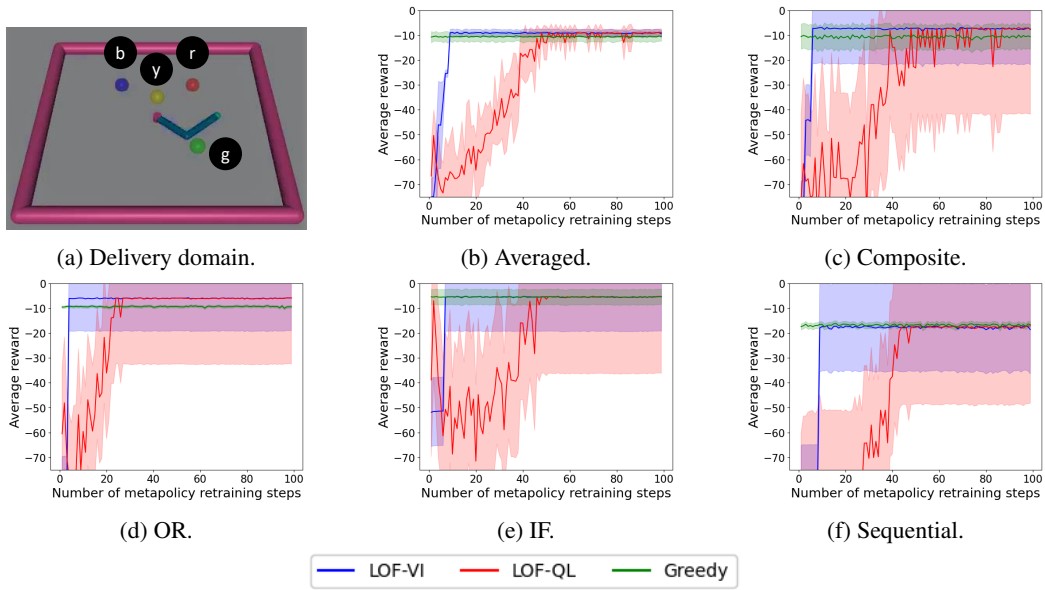

Figure 16: All composability experiments for the reacher domain.

to the FSA will alter the resulting policy in a direct and predictable way. Therefore, for example, if an incorrect human-specified task yields undesirable behavior, with our framework it is possible to tweak the task and test the new policy without any additional low-level training (however, tweaking the safety rules would require retraining the logical options).

**What happens if there is a rule conflict?**    If the specified LTL formula is invalid, the LTL-to-automaton translation tool will either throw an error or return a trivial single-state automaton that is not an accepting state. Rollouts would terminate immediately.

**What happens if the agent can't satisfy a task without violating a rule?**    The solution to this problem depends on the user's priorities. In our formulation, we have assigned finite costs to rule violations and an infinite cost to not satisfying the task (see Appendix B). We have prioritized task satisfaction over safety satisfaction. However, it is possible to flip the priorities around by terminating training/rollouts if there is a safety violation. In our proofs, we have assumed that the agent can reach every subgoal from any state, implying either that it is always possible to avoid safety violations or that safety violations are allowed.

**Why is the safety property not composable?**    The safety property is not composable because we allow safety propositions to be associated with more than one state in the environment (unlike subgoals). The fact that there can be multiple instances of a safety proposition in the environment means that it is impossible to guarantee that a new option policy will be optimal if retraining is done only at the level of the safety automaton and not also over the low-level states. In order to guarantee optimality, retraining would have to be done over both the high and low levels (the safety automaton and the environment). Our definition of composability involves only replanning over the high level of the FSA. Therefore, safety properties are not composable. Furthermore, rewards/costs of the safety property can be associated with propositions and not just with states (as with the liveness property). This is because a safety violation via one safety proposition (e.g., a car going onto the wrong side of the road) may incur a different penalty than a violation via a different proposition (a car going off the road). The propositions are associated with low-level states of the environment. Therefore any retraining would have to involve retraining at both the high and low levels, once again violating our definition of composability.

**Simplifying the option transition model:**    In our experiments, we simplify the transition model by setting $\gamma = 1$, an assumption that does not affect convergence to optimality. In the case where

$\gamma = 1$, Eq. 2 reduces to $T_o(s'|s) = \sum_k p(s', k)$. Assuming that the option terminates only at state $s_g$, then Eq.2 further reduces to $T_o(s_g|s) = 1$ and $T_o(s'|s) = 0$ for all other $s' \neq s_g$. Therefore no learning is required for the transition model. For cases where the assumption that $\gamma = 1$ does not apply, (Abel & Winder, 2019) contains an interesting discussion.

**Learning the option reward model:**  The option reward model $R_o(s)$ is the expected reward of carrying out option $o$ to termination from state $s$. It is equivalent to a value function. Therefore, it is convenient if the policy-learning algorithm used to learn the options learns a value function as well as a policy (e.g., Q-learning and PPO). However, as long as the expected return can be computed between pairs of states, it is not necessary to learn a complete value function. This is because during Logical Value Iteration, the reward model is only queried at discrete points in the state space (typically corresponding to the initial state and the subgoals). So as long as expected returns between the initial state and subgoals can be computed, Logical Value Iteration will work.

**Why is `LOF-VI` so much more efficient than the `RM` baseline?**  In short, `LOF-VI` is more efficient than `RM` because `LOF-VI` has a dense reward function during training and `RM` has a sparse reward function. During training, `LOF-VI` trains the options independently and rewards the agent for reaching the subgoals associated with the options. This is in effect a dense reward function. The generic reward function for `RM` only rewards the agent for reaching the goal state. There are no other high-level rewards to guide the agent through the task. This is a very sparse reward that results in less efficient training. `RM`'s reward function could easily be made dense by rewarding every transition of the automaton. In this case, `RM` would probably train as efficiently as `LOF-VI`. However, imagine an FSA with two paths to the goal state. One path has only 1 transition but has much lower low-level cost, and one path has 20 transitions and a much higher low-level cost. `RM` might learn to prefer the reward-heavy 20-transition path rather than the reward-light 1-transition path, even if the 1-transition path results in a lower low-level cost. In theory it might be possible to design an `RM` reward function that adjusts the automaton transition reward depending on the length of the path that the state is in, but this would not be a trivial task when accounting for branching and merging paths. We therefore decided that it would be a fairer comparison to use a trivial `RM` reward function, just as we use a trivial reward function for the LOF baselines. However, we were careful to not list increased efficiency in our list of contributions; although increased efficiency was an observed side effect of LOF, LOF is not inherently more efficient than other algorithms besides the fact that it automatically imposes a dense reward on reaching subgoals.

