# OpenReview forum: "The Logical Options Framework"
_ICLR.cc/2021/Conference — Reject_

### Official Review · AnonReviewer2 · 2020-10-28
**The paper presents a good idea towards using RL to learn sub-policies which can be composed to accomplish high-level tasks specified by Finite State Automata (FSA). However, the technical components of the paper need be clarified, and in some places fixed.**

**Rating:** 4
**Confidence:** 5

**Review:**

Summary of the work:
The authors propose the Logical Options Framework (LOF) --- a framework for reasoning over high-level plans and learning low-level control policies. This framework uses Linear Temporal Logic (LTL) to specify properties (high-level tasks) in terms of propositions. The authors propose a framework in which a separate sub-task policy is learnt to accomplish each such sub-task proposition. These low-level control policies may be reused, without further training, to accomplish new high-level tasks by performing value-iteration in the proposed Hierarchical SMDP. Experimental results demonstrate the method’s effectiveness for several tasks and experimental domains.

Quality and Clarity:
The paper does a good job at intuitively describing the value that the proposed “LOF” framework provides (satisfaction, optimality, and composability). In general, the language used in the paper is clear and easy to read.
However, some of the technical aspects in the paper are difficult to follow concretely. There are various vague statements and definitions throughout the paper. Typos, inconsistencies, and missing explanations/discussion of seemingly important notions make some of the presented ideas imprecise. This raised several questions from me on the general applicability of the framework (as it is presented in this paper) beyond the presented experimental tasks, and on some of the specifics of Theorem 3.1. Given that the focus of this paper is on the development a new framework for RL, fixing these issues is crucial. I have included a more detailed list of feedback for the authors at the end of the review.

Originality and Significance:
The proposed method defines “logical options” – sub-task policies whose goal is to trigger propositions that cause transitions in the automata-based description of the high-level task. Temporally extended tasks may then be accomplished by deploying the appropriate sub-task policy at the appropriate FSA state. This idea seems very similar to learning with Reward Machines. This similarity is acknowledged in the paper. The main conceptual difference between LOF and RMs, is that LOF proposes to learn policies that trigger transitions in an automaton, whereas learning with RMs instead learns separate policies for each state of the automaton. This difference means that in the LOF framework, previously learnt sub-task policies can be reused by composing them to complete new high-level tasks. This is not true for RMs. However, the policies learnt by LOF are not guaranteed to be optimal unless certain conditions are met.

This is a very good general idea, and as far as I am aware other works have not studied how one might compose previously learned sub-task policies to accomplish new tasks described by automata in this way. However, I am concerned that the significance of the work might be limited to the specific examples of the paper.

For example, by associating a cost with each safety proposition, instead of forming the safety FSA associated with the safety property, it seems that the only safety properties that can be expressed by  LOF are those of the form: “avoid these states”. Because this is a significant limitation in comparison with all possible safety properties, an explicit discussion of the safety properties that can be represented by LOF would help the reader to understand exactly what problems LOF can solve.

Furthermore, it is unclear how the composability results change in the presence of new and/or changed safety properties. I elaborate further on this point at the end of the next section of the review.

Questions surrounding theorem 3.1
The main theoretical result of this paper is that under appropriate conditions, the hierarchical policy learned through the LOF framework will be equivalent to the optimal policy (optimal when planning is allowed with respect to low-level actions in the FSA-MDP product).

The proof follows the logic that given the appropriate reward functions, the option policies will always reach the states corresponding to their sub-goals, and that the meta-policy will select a sequence of sub-goals that always reaches the FSA goal state.
The proposed method assumes that if an option corresponding to a particular sub-goal is selected by the meta-policy, then no other sub-goal proposition will be triggered before that option is complete. My concern with this assumption is as follows: what happens in scenarios in which an option policy passes through a state associated with a different sub-goal before reaching its own sub-goal? Would it then be possible for the meta-policy to select an option corresponding to a particular sub-goal, but for a different sub-goal proposition to be triggered first, causing an unwanted transition in the FSA?

For example, suppose in the delivery domain we could complete a task by either of the following sequences of sub-goals: “a then b”, or “b then a then c”. If “a then b” is less costly than “b then a then c”, the optimal meta-policy should result in “a then b”. Furthermore, assume that “b” lies directly between “a” and the initial state of the agent. The optimal policy for sub-goal proposition “a” would be to move directly to “a”, which would cause it to first pass through “b”. This would cause the agent to be forced to follow “b then a then c” even if the meta-policy first chose proposition “a”. Meanwhile, the optimal policy allowing low-level planning in the FSA-MDP product space would be to move to “a” while actively avoiding “b”, and then to proceed to “b”.

Also, it is unclear whether consideration of obstacles and safety propositions are included in the consideration of theorem 3.1. It is unclear what the reward functions corresponding to the “goal-conditioned policy” is. If R_S is included in the reward function used to train sub-task policies, then these policies will learn to avoid obstacles. When we compose these sub-task policies for a new task with potentially different safety properties, the policies we learned previously could be sub-optimal in the new scenario.

Experimental results:
The experimental results demonstrate the effectiveness of the framework for the chosen experimental tasks. I like the videos provided in the supplementary materials; they are nice visualizations of how the sub-policies are strung together to complete the overall task.

From the paper’s description, it is unclear how LOF-QL is implemented. How does this algorithm make use of the liveness FSA if it does not have access to the FSA’s transition function?

Also, I am concerned that the comparison against RMs may not be fair. As stated in the paper, the RM-based method is only rewarded when the final RM state has been reached. Conversely, the reward for the LOF method is much denser. I do not see why the RM-based method could not also be rewarded for each “good” transition.

Finally, the experiments include the “can” proposition which represents when one of the sub-goals has been canceled. It is unclear how the labeling function, which is defined to map specific MDP states to individual propositions, could return this proposition. If the proposition is returned randomly, the formalisms of the labeling function need to be altered to include these types of randomly occurring propositions.

Pros

> The general idea of the paper is good. I like the idea of pre-learning sub-task policies, and then of using automata-based task descriptions to find meta-policies that accomplish new tasks without additional training. This is the “composability” described in the paper.

>The authors do a good job of intuitively describing the benefits that this type of method could provide over competing algorithms.

>The experimental tasks (particularly the videos of the tasks being completed) provide a nice visualization/demonstration of the ideas of the paper.

Cons

>The technical aspects of the paper are hard to follow concretely.

>Some of the theory seems vague and the paper has various typos and inconsistencies. This could lead to reader misinterpretations and/or author mistakes.

>The method’s applicability seems to be somewhat limited to specific types of tasks, without explicit discussion of exactly what types of tasks can be solved.

>The questions raised surrounding theorem 3.1 need to be clarified.

Further detailed feedback:

>How are the logical options learned? In particular, how are T_{o_p}(s’|s) and R_{o_p}(s) learned. These are the components of the options that are subsequently used to find the metapolicy. Are they estimated by averaging values over rollouts of the learned policy? What if the environment is highly stochastic and their values vary greatly across different rollouts?

>The assignment for T_{o_p}(s’|s) on line 11 of algorithm 1 seems to assume there is a fixed k in which p will be reached. Would this always be the case if the environment is stochastic? This also seems to be a different definition than is given in equation 2.

>In equation 1, R_{o}(s) is defined as the expected cumulative reward of options given that the option is initiated in state s at time t and ends after k time steps; shouldn’t this make R_{o}(s) a function of k as well?

>In the definition of T_{o}(s’|s), what is p(s’,k)? I assume this is the probability, under the option’s policy, of reaching state s’ after k timesteps from state s. If this is the case, p(s’,k) should be defined.

>In line 9 of the Algorithm 1, T_{P}(s,p) = 1 is written. On line 11 of Algorithm 1, T_{P}(s) = p is written instead, but both have the same meaning: proposition p is returned by the labeling function from state s.

>The liveness FSA’s transition function is defined and treated as a probability distribution. However, the automaton’s transitions appear to be deterministic in the presented examples. If there is a particular reason for it to be defined as a transition distribution, some discussion would be helpful for the reader.

>In the definition of a Hierarchical SMDP, the transition function is defined as a cartesian product of the environment transition distribution, the labeling function, and the FSA transition function. The precise meaning of this notation is unclear. A brief description of how a transition proceeds would be very helpful for the reader to understand the order in which arguments are passed to these three functions. Does the environment transition first, and the proposition of the new environment state cause the FSA to transition?

>In section 3.1 and in appendix A.1, the author writes that “if the optimal option policy equals the goal-conditioned policy, the policy will always reach the sub-goal”. This will not be the case if the environment is stochastic and it is possible for the agent, under ANY policy, to slip into a state from which the sub-goal is not reachable.

>Point 2 of the contributions outlined in section 1.1 states that the options can be composed to solve new tasks with only 10-50 retraining steps. This wording is at odds with the abstract, which states that LOF’s learned policies can be composed to satisfy unseen tasks with no additional learning.

>In Section 3 “These may only appear in the liveness property and there may only be one instance of each subgoal.” The second half of this sentence is unclear. Does this mean that the inverse image of each sub-goal proposition through the labeling function is a singleton set containing only one state?

>In section 3 “Safety propositions are propositions such as ‘the agent received a call’ that the agent has no control over or does not seek out.” This sentence is unclear. It would help if you could express what it means for the agent to “have control over” or to “seek out” a proposition in terms of the MDP states, actions, and proposition labeling function.

---

> ### Author Response · Authors · 2020-11-19
> **Response to AnonReviewer2 [part 1]**
>
> We thank the reviewer for such a thoughtful and in-depth review. This review is really valuable to us and we have taken the feedback to heart. We believe that the reviewer has pointed out two major issues with our formulation that we have addressed in the revision. 1) The fact that our formulation in the paper does not cover the case where the safety property is non-trivial and the case where the LTL specification can’t be separated into independent liveness and safety specifications (discussed in points 1 and 3 below). We have added a new section to the appendix (Appendix A) where we present a formulation that is much more general than the one presented in the paper. 2) The fact that our formulation of the proposition labeling function is inconsistent with randomly occurring safety propositions (discussed in point 7 below). We added a new category of propositions called event propositions to address this concern. Furthermore, the reviewer has pointed out numerous other issues that we have attempted to address. We apologize for the length of this review and the fact that it has been split into so many comments, but the reviewer pointed out many really good and deep points that have taken a lot of space to address.
>
> 1) “it seems that the only safety properties that can be expressed by LOF are those of the form: “avoid these states”” – LOF is not limited to these types of safety properties. In the submission, we briefly mention this when we say, “In our experiments, the safety property $\phi_{safety}$ defines a reward function R_S that penalizes violations of safety rules. $\phi_{safety}$ can be converted into an FSA $\mathcal{T}_S$, and the reward function can be defined on the states of $\mathcal{T}_S$. However, to limit the complexity of the formulation we assume that there is simply a reward (cost) associated with every safety proposition.” However, we think that your concern should be addressed properly, so we have now included a more general formulation of LOF in Appendix A that can handle any safety property. We want to emphasize that LOF can handle any type of safety property, as long as the low-level policy-learning algorithm can accommodate it. In the general formulation, the safety property is translated into an automaton, and Reward Machines or other appropriate algorithms can be used to learn logical options that obey the safety property.
>
> 2) “it is unclear how the composability results change in the presence of new and/or changed safety properties”: Safety properties are not composable. In our definition of composability, we assume that the liveness property can change but not the safety property. We were not clear about this in the paper; in our definition of composability in the introduction, we refer to “tasks” but should use the more specific term “liveness property.” We will edit the revision to be very clear about this.

---

> ### Author Response · Authors · 2020-11-19
> **Response to AnonReviewer2 [part2]**
>
> 3) “scenarios in which an option policy passes through a state associated with a different sub-goal before reaching its own sub-goal” – this is a very interesting scenario that really dives into the nuances of this framework. But we want to stress that this scenario is fully compatible with LOF. There are two ways to interpret the example specification that you give, “’a then b’, or ‘b then a then c’”. The first way is to interpret the specification in LTL as F(a & Fb) | F(b & F(a & Fc)) (to see the resulting automaton you can input this specification into the web app at https://spot.lrde.epita.fr/app/). In this case, even if the meta-policy chooses to go to a but ‘accidentally’ encounters b, this won’t affect the execution of the metapolicy. Encountering b will not change the FSA state of the agent, and the agent will continue to head towards option a since that is optimal.
>
>  However, we suspect that you mean something along the lines of, “’a then b’, or ‘b then a then c’, and either b cannot be true until a is true or a cannot be true until b is true.” In LTL, this can be stated as (G(!b U a) & F(a & Fb)) | (G(!a U b) & F(b & F(a & Fc))). In this scenario, if the agent ‘accidentally’ goes to goal b, then it will have committed itself to the suboptimal “b then a then c” path. This specification is different from the previous one in a significant way. Note that G(!b U a) and G(!a U b) can be thought of as safety properties. But we have violated one of LOF’s assumptions, which is that subgoals cannot appear in safety properties. This is what causes the paradox in your example – in the case where the agent has decided that it wants to go to goal a, thereby committing itself to the “a then b” path, goal b is actually not a subgoal but a safety proposition (something to be avoided) until the agent reaches goal a. We can bring this specification into line with LOF’s requirements by making new safety properties, let’s call them “sa” and “sb”. “sa” coincides with goal a, and “sb” with goal b. We can then write a new specification, (G(!sb U sa) & F(a & Fb)) | (G(!sa U sb) & F(b & F(a & Fc))). Although this specification cannot be written as separate liveness and safety properties, it can be translated into a Buchi automaton, and it is always possible to decompose a Buchi automaton into liveness and safety properties [1]. We skimmed over the possibility that the LTL specification can’t be written in terms of independent liveness and safety properties, but we have significantly revised the paper to take this into account because it is a much more general scenario. This more general formulation is in Appendix A. In this situation, the logical options can be trained with an algorithm such as RM. The safety property is not composable, so using an optimal and satisfying, but not composable, algorithm to learn the logical options makes sense. (We have also added a discussion of why the safety property is not composable in Appendix D). Taking the safety automaton and assigning costs to the states, the logical option for achieving goal a would follow rules along the lines of “Avoid sa unless sb has been visited or avoid sb unless sa has been visited.” This would resolve the paradox of your example and enable the agent to perform an optimal trajectory.
>
>  We imagine that the scenario that you have described would be a major stumbling point in any algorithm that aims for composability. But we hope we have convinced you that LOF provides a sound framework for reasoning through this example – particularly the usefulness of the expressive precision of LTL specifications and the way in which LOF’s distinction between liveness and safety properties and subgoal and safety propositions was necessary in order to resolve the paradox. However, your example has also shown us the importance of covering the case where the LTL formula cannot be divided into independent liveness and safety properties (now covered in Appendix A). In this more general case, the LTL formula would have to be translated into a Buchi automaton and then factored into a liveness automaton and a safety automaton. Any subgoal propositions occurring in the safety automaton can be dealt with by creating a “safety twin” for that proposition (for example, “sa” for “a” and “sb” for “b”) and using the “safety twin” instead of the subgoal in the safety automaton (we have added a discussion of “safety twins” in Appendix A.1). An algorithm such as RM could then be used to learn logical options that obey the safety automaton.

---

> ### Author Response · Authors · 2020-11-19
> **Respone to AnonReviewer2 [part3]**
>
> 4) “it is unclear whether consideration of obstacles and safety propositions are included in the consideration of theorem 3.1” – they are considered implicitly in terms of the reward and value functions, and on our assumption that the subgoals are reachable from every state in the environment. We will try to make this more clear. “It is unclear what the reward functions corresponding to the “goal-conditioned policy” is” – we have added a more formal definition of the goal-conditioned policy to the revision in Appendix B (the proof section). The reward function of the goal-conditioned policy is $R_\mathcal{E}$, the reward function of the environment, which includes $R_S$. However, the objective of the goal-conditioned policy is not to maximize $R_\mathcal{E}$ but rather to maximize it with the constraint that the goal is reached in a finite amount of time. And yes, new tasks cannot have new safety properties. When defining composability in the Introduction, we say that composability is the ability for the framework to very quickly learn new metapolicies for new tasks. We said “tasks” instead of “liveness properties” because we had not yet defined that term. We have revised the definition of composability in the Introduction to be more clear about this.
> 5) “it is unclear how LOF-QL is implemented”: We have added a detailed description of LOF-QL in Appendix C.3. LOF-QL uses Q-Learning instead of Value Iteration to find a policy over the FSA. It does this in the normal Q-Learning way; it starts at the initial state of the FSA and samples from the options; it then transitions to the next state and samples again. As it does this it builds Q table which is used as the policy. We mention in the paper that the transition function $T_F$ does not need to be known to do Q-Learning – we mean that it does not need to be known explicitly, as Q-Learning relies on sampling the environment rather than using the transition function explicitly as in value iteration. We have changed our wording in the paper to reflect this.
> 6) “I am concerned that the comparison against RMs may not be fair”: We debated about how to design the RM reward function. If we made the reward function dense (by rewarding every state transition), it would make learning much more efficient and probably the same as LOF in some cases. However, imagine an FSA with two paths to the goal state, “a -> b” and “c -> d -> e -> f”.  If state transitions are rewarded, RM might learn to prefer the reward-heavy 4-goal path rather than the reward-light 2-goal path, even if the 2-goal path results in a lower low-level cost. In theory it might be possible to design an RM reward function that adjusts the automaton transition reward depending on the length of the path that the state is in, but this would not be a trivial task when accounting for branching and merging paths. We therefore decided that it would be a fairer comparison to use a trivial RM reward function, just as we use a trivial reward function for LOF. However, we were careful to not list increased efficiency in our list of contributions; although increased efficiency was an observed side effect of LOF, LOF is not inherently more efficient than other algorithms besides the fact that it automatically imposes a dense reward on reaching subgoals. We have included this discussion in Appendix D.
> 7) “If the proposition is returned randomly, the formalisms of the labeling function need to be altered to include these types of randomly occurring propositions”: Thank you for pointing out this shortcoming in our formulation. We initially viewed the “can” proposition as kind of a bonus ability that we tacked on to the formulation, but you’ve made us realize that this needs to be given the full formal treatment. We therefore reflected more on our formulation, and we decided to split “safety propositions” into two types – true safety propositions that appear only in the safety property, and “event” propositions that can appear in the liveness property (in the general formulation of Appendix A, they can also appear in the safety property). Event propositions are not associated with low-level states of the environment, and their values are set at the beginning of the rollout. Therefore their values can vary and affect how the task is executed, but they are known. They must be known from the beginning of the rollout because if they were not known, then we would be dealing with POMDPs and POMDP planning, which would go beyond the scope of our paper. However, even if the values of the event propositions are not known immediately and only revealed to the agent later, satisfaction and composability are still guaranteed, just not optimality. We have updated the formulation in Section 3 and Appendix A to include event propositions.

---

> ### Author Response · Authors · 2020-11-19
> **Response to AnonReviewer2 [part4]**
>
> Responses to detailed feedback:
>
> 1, 2, 4) We regret that we weren’t able to fit a deeper discussion of $T_{o_p}(s’ \vert s)$ and $R_{o_p}(s)$ in the paper, but we have attempted to include more discussion of them in Sections 2 and 3 and in Appendix D of the revision. Let us refer to the transition model as $T_o$. For $T_o$, we define the general equation in Eq 2 (using the terminology from the options framework paper [1]). In logical options, we assume that the option can only terminate at a single state, the subgoal associated with the option. Let’s called this state s_g. As you guess in feedback #4, $p(s’, k)$ is the probability of the option terminating at state s’ after k timesteps. We have added this definition in the paper. We can use this knowledge to simplify the $p(s’, k)$ term in Eq 2, since $p(s’, k) = 0$ for all $s’ \neq s_g$.
>
>  Next, you point out that we treat k as a deterministic variable. This was an error and we have updated the formulation to account for k being a random variable.
>
>  In our experiments we simplify the transition model by setting $\gamma = 1$. In the case where $\gamma = 1$, Eq 2 reduces to $T_o(s’ \vert s) = \sum_k p(s’, k)$. Assuming that the option terminates only at state $s_g$, then Eq 2 further reduces to $T_o(s_g \vert s) = 1$ and $T_o(s’ | s) = 0$ for all other $s’ \neq s_g$. Therefore, no learning is required for the transition model. However, your point remains valid, that learning the transition model of options is one of the biggest challenges of using the option framework. However, this has not prevented the options framework from being a popular HRL frameworks. Our assumption that the subgoal is associated with a single state greatly simplifies the transition model (to being a matter of learning the distribution over number of time steps), but even this problem is challenging. [2] discusses how to address this problem.
>
>  The reward model $R_{o_p}(s)$ is equivalent to a value function (it is the expected return of carrying out the option to termination). Therefore, it is convenient if the policy-learning algorithm used to learn the options learns a value function as well as a policy (e.g., Q-learning and PPO). However, as long as the expected return can be computed between pairs of states, it is not necessary to learn a complete value function. This is because during Logical Value Iteration, the reward model is only queried at discrete points in the state space (typically corresponding to the initial state and the subgoals). So as long as expected returns between the initial state and subgoals can be computed, Logical Value Iteration will work. We have included this discussion in Appendix D.
>
> 3\) “shouldn’t $R_{o}(s)$ be a function of k as well?”: Yes, $R_o(s)$ is a function of k. We have edited Eq 1 to make this more clear. Our notation for $R_o(s)$ in Alg. 1 line 12 was also confusing, and we have changed it to include k and hopefully be less confusing.
>
> 5\) “In line 9 of the Algorithm 1, T_{P}(s,p) = 1 is written”: Thank you for pointing out this inconsistency in our notation. We have fixed this.
>
> 6\) “The liveness FSA’s transition function is defined and treated as a probability distribution”: Thanks for pointing this out. We define the FSA transition function as a probability distribution in keeping with common notational practices. We have added a sentence to Sec. 3/Hierarchical SMDP/paragraph 2 to make this more clear.
>
> 7\) “Does the environment transition first, and the proposition of the new environment state cause the FSA to transition?”: This is an important point that we spent a lot of time thinking about, as we had to implement it in the code, but it seems like it didn’t make it into the paper. The order is environment transition function; proposition labeling functions; FSA transition function. We will definitely add this to the revision. The reason for this order is because when the agent executes an action, it first has to affect the environment. The updated environment state is then used to find the active proposition, which is then used to find the next FSA state. We have updated the revision to mention this (in Sec. 3/ Hierarchical MDP/paragraph 4).

---

> ### Author Response · Authors · 2020-11-19
> **Response to AnonReviewer2 [part5]**
>
> 8) “it is possible for the agent, under ANY policy, to slip into a state from which the sub-goal is not reachable”: This is a good point that deserves more discussion. In the RL/MDP literature, it is standard to make a regularity assumption that every state is reachable from every other state. We make a similar assumption that the subgoal can be reached from any point in the state space, so it should always be possible for the agent to reach the subgoal, even in the case of stochastic transitions. (We noticed that we did not state this assumption in the proof in Appendix B, so we have added it there). In the case of an environment with stochastic transitions where there are states from which the subgoal is unreachable, it is impossible to guarantee that the agent will reach the subgoal/satisfy the specification, not just for our algorithm but for any algorithm. It is probably possible to bound and maximize the probability of satisfaction, but that is beyond the scope of our current work.
>
> 9) “Point 2 of the contributions outlined in section 1.1 states that the options can be composed to solve new tasks with only 10-50 retraining steps. This wording is at odds with the abstract”: We apologize for the confusion about our terminology regarding retraining. In the abstract we wanted to avoid direct reference to the experiments – by “no additional learning”, we meant “no additional sampling of the environment.” But we don’t want to mislead readers, and it would be more clear to say “10-50 retraining steps.” We have changed the abstract.
>
> 10) ““These may only appear in the liveness property and there may only be one instance of each subgoal.” The second half of this sentence is unclear”: Yes, each subgoal may be associated with only one state. We have clarified this in the revision as “Each subgoal may only be associated with one state.”
>
> 11) “It would help if you could express what it means for the agent to “have control over” or to “seek out” a proposition in terms of the MDP states, actions, and proposition labeling function”: This is a good point. We have revised our definitions of subgoal, safety, and event propositions in both Sec. 3 and Appendix A. Subgoals appear only in the liveness property and may only be associated with one low-level state. In order to satisfy the liveness property, the agent must traverse transitions through the liveness property to reach the goal state. The only way the agent can do this is by seeking out subgoals. We assume that the goal state is reachable from every state of the liveness property using only subgoals. Safety propositions appear only in the safety property. Every state of the safety property is an accepting state, and violations occur when undefined/unallowed transitions occur. Therefore, propositions that cause unallowed transitions must be avoided. Event propositions appear in the liveness property (in the general formulation in Appendix A, they may also occur in the safety property). Since event proposition truth values are set by the environment at the beginning of the rollout, the agent does not “have control over” event propositions.
>
> [1] https://www.cs.cornell.edu/fbs/publications/RecSafeLive.pdf
>
> [2] https://www.ijcai.org/Proceedings/2019/270
>
> [3] https://people.cs.umass.edu/~barto/courses/cs687/Sutton-Precup-Singh-AIJ99.pdf

---

### Official Review · AnonReviewer3 · 2020-10-29

**Rating:** 6
**Confidence:** 2

**Review:**

In this paper, the author proposes logic option framework for learning policies that satisfy the logic  constraints.  Temporal logic rules are converted into finite state machine and each of them is then converted to a learnable option, where the logic proposition is used as the reward function. Given the options, it searches for the optimal meta-policy over all the options. The LOF is tested on a gridworld and an OpenAI Gym benchmark and is compared against baselines that do not use rules.

I find the idea of integrating logics into RL is an exciting direction for improving agent's interpretability and the proposed LOF does have some novelties in applying the FOL rules. However, I'm not familiar with the RL and option literature and it's difficult for me to tell its original work from the existing ones on the RL side.

Some questions for the author:

Judging from Algorithm 1, it seems the rules and predicates are pre-defined with the environment and their conversion into FSA and reward functions are also done with an existing method. Is there anything original in terms of rule/predicate learning or conversion for the proposed LOF?

In Algorithm 1, each rule is converted to an option and learned independently. What does the LOF do if rules are correlated or have conflicts?

- Say, rule1 is a pre-requisite of rule2: rule1: "go to dest1" and rule2:"if agent is in dest1 then go to dest2, otherwise check rule1"

- Or rule1: "go to dest1 asap" whereas rule2: "do not violate the traffic rules"

- Does the LOF provide mechanism to address the above issues?

I'm also concerned about the scalability of this method. It seems to me that the options need to be evaluated at each possible state in the environment. If the goals are fixed, probably one can store the value beforehand, but for a dynamic environment where the goal is changing, how much does it cost to update the reward function and learn the option policies? And how well does it scale with the size of the state/action space?

---

> ### Author Response · Authors · 2020-11-19
> **Response to AnonReviewer3 [part 1]**
>
> We are grateful to the reviewer for the questions and constructive feedback, and hope that we can clarify some of aspects of our work.
> 1) Most of the LTL+RL literature achieves satisfaction and (sometimes) optimality by using the automaton associated with the LTL specification to shape the reward function, including Reward Machines. During training, the environment keeps track of what state the automaton is in and rewards the agent whenever a (favorable) transition is achieved. This approach can guarantee satisfaction and optimality; however, it is inherently not composable because the automaton rewards get baked into the learned value function/policy. If the automaton is changed, policy learning typically has to be done from scratch (we have added Appendix Figures 4, 5, 6, and 7 to illustrate how LOF differs from Reward Machines, the published work most similar to ours). Other work in symbolic planning is more similar to ours in that they learn high- and low-level controllers. However, these works do not take into account the specific tradeoffs between composability and optimality that we cover in our paper, and their high-level controllers are either not composable or only capable of limited composability. We will try to emphasize the differences between our work and the related work more in the revision.
>
>  To address your question of if there is “anything original in terms of rule/predicate learning or conversion for the proposed LOF”: we assume that all aspects of the hierarchical semi-MDP are known, including the rules and propositions. Learning rules and propositions is a very exciting goal and definitely an avenue of future work; in this paper, we have laid the groundwork for that type of future work by defining the formulation of the hierarchical SMDP and proposing an algorithm for finding a satisfying/optimal/composable policy over it. Our hierarchical SMDP formulation is unique in that it includes both a hierarchical action space (as in the original options framework paper) as well as a hierarchical state space/transition function/reward function (which is common to many LTL+RL papers).
>
> 2) Rule conflicts are a topic that we didn’t have space to address in the initial submission, but we will talk about it in the revision. Can you elaborate more about your first example? As stated, the rules will cause the agent to go to dest1 and then dest2. But in general, if rules are logically incompatible, they will probably translate into a single-state automaton that is not an accepting state. For example, the formula “(!a U b) & (!b U a) & G!(a&b) & F a” can be thought of as, “Go to A, but you can’t go to A until you’ve gone to B, and you can’t go to B until you’ve gone to A”. It results in a single-state automaton with no goal state. (You can try inputting this formula into https://spot.lrde.epita.fr/app/ to see for yourself). LOF is compatible with such an automaton; the policy will simply terminate at the first time step. This is more an issue of user error; hopefully the user will see that the formula does not make logical sense (perhaps upon seeing that the automaton is trivial) and correct the formula. We have included this discussion in Appendix D, “Further Discussion”.
>
>  You make a good point with the second example – what if achieving the goal involves violating a safety constraint? One of our assumptions is that the agent can achieve any subgoal from any state of the environment, so an environment that requires a violation of a hard safety constraint would violate our assumption. We believe this is a reasonable assumption to make because it is not fair to the agent to require the agent to achieve a task that is against the rules. However, under most conditions LOF will still “work” in such an environment. For example, in the formulation that we use in the paper, safety propositions are assigned negative but finite costs. As we discuss in the proofs of satisfaction/optimality, the cost of not satisfying the task is negative infinity, whereas the cost of violating the rules is negative but finite. Therefore, in our specific formulation, if the agent is in an environment where violating the safety rules is the only way to achieve the task, it will learn to violate the safety rules at the cost of a massive negative reward (however, it will only learn this behavior in the limit of an infinite number of training steps; limiting the length of a training episode to a reasonable number will prevent the agent from ever violating safety rules with large negative costs). We have included this discussion in Appendix D, “Further Discussion”.
>
>  We hope that we have explained how these issues can be handled and how LOF would react to them. We are happy to clarify if you have more questions.

---

> ### Author Response · Authors · 2020-11-19
> **Response to AnonReviewer3 [part 2]**
>
> 3) With regard to scalability, these methods are as scalable as whatever low-level algorithm is used to train the logical options (in our case we used Q-Learning and PPO, but there is almost no limit to what low-level algorithm can be used). Therefore the size of the state/action space is not too much of a problem.
>
>  One challenge, as you point out, is that LOF must learn a value function for every option (the “option reward model”). Fortunately, therefore are many scalable deep RL methods that approximate value functions (such as PPO). Moreover, in practice, it is not necessary for the value function to be well-known at every point in the state space. This is because the value function is only queried wherever the agent needs to select a new option, and this typically occurs at a small subset of points in the state space (for example, at the initial state and at the subgoals).
>
>  The situation becomes much more difficult in the case of a dynamic environment, but it is still not insurmountable. The challenge is that a time dimension has to be added to the state space, which makes the state space much larger. In addition, our assumption that subgoals are associated with a single state would have to be relaxed so that subgoals can be associated with a single state at an arbitrary time. Moving subgoals make the problem even harder and force an even greater relaxation of our assumption so that at any given point in time, the subgoal is associated with a single state (but this state may change over time). However, as long as it is possible for the logical options to fulfill the condition that the subgoal can be reached from any point in the state space (and keeping in mind that almost any low-level RL algorithm can be used to learn the options), it should be possible to learn satisfying and composable policies using LOF (although possibly not optimal, as the condition on associating the subgoal with a single state would have to be relaxed).

---

### Official Review · AnonReviewer1 · 2020-10-30
**Interesting work with good results, but a bit incremental to the literature**

**Rating:** 6
**Confidence:** 4

**Review:**

This paper is on a new RL framework that leverages logical reasoning to improve the learning performance of RL agents. In particular, the knowledge is encoded using LTL, and includes both safety knowledge (used for reward function definition) and liveness knowledge (used for constructing FSA). THe developed framework has been evaluated using tasks in both discrete and continuous domains, where the RL agent was realized using Q-learning and PPO respectively. The framework was compared with baselines including another LTL-based RL methods (Reward Machines). Results show that LOF performed better than the baseline methods in learning rate and policy quality (in most cases).

The whole idea of leveraging human knowledge to improve RL agent's learning performance makes senses. The main concern is that the developed framework looks quite incremental in comparison to the methods from the literature. For instance, there are already methods in the literature using LTL to bias the reward function, e.g., the papers that improved the Reward Machines work. As a result, the novelty of the safety knowledge is rather incremental. At the same time, the "liveness" part is quite similar to those methods that plan at a symbolic level to guide RL agents, such as the following:

Lyu D, Yang F, Liu B, Gustafson S. SDRL: interpretable and data-efficient deep reinforcement learning leveraging symbolic planning. 2019

It's nice to see the discussions on satisfaction, optimality, and composability. The analysis on optimality is the most important, whereas the other two points are relatively trivial due to the LTL-based FSA. But it turned out the developed approach, LOF, does not guarantee any desired properties beyond those from the literature. In particular, the LTL-based FSA cannot guarantee global optimality, where the baseline of Reward Machines does.

The reviewer is curious about if Reward Machines can be considered an ablation of LOF, where the "liveness" component is disabled (and "safety" is retained). If not, what are the differences between the two methods?

Knowledge-based RL can go beyond HRL methods. For instance, the following survey paper summarizes a few other ways of leveraging human knowledge in RL. Some discussions can help improve the related work section.

Zhang S, Sridharan M. A Survey of Knowledge-based Sequential Decision Making under Uncertainty. 2020

In general, it's not surprising to see human knowledge is useful in RL. However, human knowledge is not always correct. It will be nice to evaluate the situations where human knowledge in incomplete or inaccurate, or at least some discussions can be helpful.

----------
The reviewer appreciates the response in detail. The new figures (Figs 4-7) are helpful for demonstrating the differences between LOF and RM, while also demonstrating that their outputs can be similar or the same in many cases. The reconciliation between optimality and composability is a nice feature. Overall, the reviewer still feels positive on this work.

---

> ### Author Response · Authors · 2020-11-19
> **Response to AnonReviewer 1**
>
> We are grateful to the reviewer for the detailed review of our paper.
> 1) We want to emphasize that improving learning performance is not one of the goals of our method, although it can be a side effect. Rather, our goal was to achieve of satisfaction, optimality, and composability. We agree that in terms of “satisfaction” and “optimality” alone, our work is not novel, as there are many other algorithms that achieve satisfaction and optimality, including RM. However, we believe that our work makes a significant contribution in terms of achieving satisfaction and optimality along with composability. As AnonReviewer2 points out, “This is a very good general idea, and as far as I am aware other works have not studied how one might compose previously learned sub-task policies to accomplish new tasks described by automata in this way” (although they are concerned about the generality of our work; in the revisions we hope we have addressed those concerns). In order to achieve composability, we make a big shift in perspective – whereas other papers achieve satisfaction/optimality by shaping the reward function using the automaton, we instead take a planning perspective, where we find an optimal policy over the automaton. Therefore, if the automaton is changed/modified, we can find a new optimal policy by replanning over the new automaton. (This is a similar approach to that taken in SDRL and other symbolic planning methods, but it differs in that the meta-controller of SDRL does not have the property of full composability that we describe in our work. But we are grateful for the reference and will include it in the related work).
>
>  With regard to the “liveness” and “safety” properties, we believe that we have made a useful contribution in identifying these two properties as distinct components of a planning algorithm. Most other papers in the LTL+RL literature do not distinguish between liveness and safety properties and plan over a single, combined automaton. However, in this paper we point out that the safety property is mostly useful for defining the reward function, and the liveness property is useful for planning. Therefore, the safety property is incorporated into the reward function for training the logical options, and the liveness property is translated into an FSA so that an optimal policy can be found over the FSA. We believe that this distinction is a useful contribution for the RL+LTL literature (although it is widely known in the formal methods community).
>
> 2) With regard to satisfaction, optimality, and composability, we believe that our most important contribution is that we have found a way to reconcile optimality and composability, which in general are opposing goals. Composability requires the policies to be flexible in a way that makes optimality difficult to achieve (hence the existence of many optimal methods that are not composable, and many composable methods that are not optimal). Furthermore, we want to emphasize that our proof of optimality guarantees global optimality under the conditions that we state. Although these conditions are stricter than the conditions that RM requires, we strove to find the loosest possible assumptions for achieving optimality while also maintaining full composability. We point out in the paper that in continuous state spaces it is impossible to meet our condition that a subgoal be associated with a single state; however, by defining the subgoal to be a small region as we do in our paper, it is possible to achieve near-optimal policies. We acknowledge that our work will not have the final say in this discussion, as future papers may be able to loosen our assumptions, but we hope that our work can kickstart the discussion.

---

> ### Author Response · Authors · 2020-11-19
> **Response to AnonReviewer 1 [part 2]**
>
> 3) It’s an interesting point if Reward Machines can be considered an ablation of LOF. Given a safety property translated into an FSA, RM (or other existing LTL+RL algorithms) could be used to learn the low-level logical options that obey the safety property. We discuss this in more detail in Appendix A. Logical options learned using RM would be compatible with LOF. RM could also learn to satisfy the liveness property, but the resulting policy would not be composable. The reason to use LOF instead of RM to learn to satisfy the liveness property is to maintain composability, as LOF’s logical options can be easily recombined to satisfy arbitrary tasks, whereas RM’s automaton-specific subpolicies cannot be recombined. We apologize for not making the difference between these two methods more clear; this seems to have caused a lot of confusion. We have added four new figures to the appendix, Figures 4, 5, 6, and 7, which we hope will illustrate the differences between LOF and RM.
> 4) Thank you for the paper reference, it is very relevant to our work and we have included it in the revision. We agree that human knowledge has the risk of being incorrect and that this is a major problem in rule-based planning. In this paper, we tried to avoid those questions and focus simply on how to formulate and solve a hierarchical SMDP, but we think that this will be a very interesting avenue of future work. One huge benefit of using LTL/FSAs at the high level is that these forms of representing rules are much more interpretable than alternatives (such as neural nets). One of the consequences of composability is that any modifications to the FSA will alter the resulting policy in a direct and predictable way. Therefore, for example, if an incorrect human-specified task yields undesirable behavior, with our framework it is possible to tweak the task and test the new policy without any additional low-level training (however, tweaking the safety rules would require retraining the logical options). Unfortunately, due to lack of space we were not able to include this discussion in the paper. However, it is an important and interesting point, so we have included it in Appendix D, “Further Discussions”.

---

### Official Review · AnonReviewer4 · 2020-11-03
**Writing and exposition require a lot of work**

**Rating:** 4
**Confidence:** 4

**Review:**

Summary: The paper proposes a hiearchical RL framework augmented with linear temporal logic. Low-level policies are trained to reach a set of subgoals with penalty on violating safety constraints, allowing the high-level policy to adapt to different task by composing the low-level policies to reach a set of subgoals specified by an Linear Temporal Logic clause.

My first comment is on technical novelty. Although the paper states that LOF takes a step beyond RM to enable compositionality, there is no empirical comparison or evaluation. The only experiment on evaluating compositionality (c, d in Figure 2) does not compare LOF against RM.

My second major comment is on technical exposition. The paper's writing needs a lot of work, especially the method section. The method section started off by describing different concepts used by the framework, but it is unclear how these concepts constitute LOF. For example, how exactly is LTL used in LOF? The introduction mentioned that LOF provides "compositional" property, but there is not a single hint of how might each piece of LOF lead to this property. It took me a few read to understand how the low-level reward might be agnostic of the high-level task, which in turn enables fast adaptation to new task because only the high-level needs to be trained, but it is better to state that clearly.

I would suggest the authors to write one or two paragraphs at the beginning of the method to provide a high-level overview of (1) What LOF is, (2) what are different components of LOF (3) how these components interact with each other, intuitively (4) what are the technical challenges each component, or all components jointly, are trying to tackle  (e.g., enabling compositionality).

In addition, there are many mentioning of using reward function in the form of an FSA. I understand that this was already introduced in Icarte et al., but it is worth reviewing it in the main paper.

Comments on experiments:
- For compositionality experiment, why might RM and Flat not be applicable here (related to my first comment)?
- I do not see RM curve in (e)


Minor:
- "\phi_{liveness} is represented as a finite state automaton". The paper has been using FSA through except first introduced FSA. It's better to be consistent.

---

> ### Author Response · Authors · 2020-11-19
> **Response to AnonReviewer4**
>
> We thank the reviewer for the thoughtful review of the paper. The reviewer has highlighted several shortcomings of our exposition that we will try to address.
> 1) We don’t compare LOF against RM in the composability experiments because RM is not composable. RM is not composable because it learns unique subpolicies for each state of the automaton. These state-specific subpolicies are also automaton-specific; for example, if the automaton is “go to A, then B”, the subpolicy for the initial state will learn to go specifically to subgoal A. If the automaton is modified to be “go to B, then A”, the initial state’s subpolicy will still attempt to go to subgoal A. There is no way to adjust RM to be composable without significant changes to the algorithm. Since RM is inherently not composable, we did not include it in the composability experiments. We have added more of an explanation in the paper, and we have also added 3 new figures to the Appendix, Figures 4, 5, 6, and 7 which we hope will illustrate how LOF differs from RM and why RM is not composable.
>
>  The “Flat Options” baseline, which represents the standard options framework, is also not composable because the baseline has no knowledge of the high-level FSA. This is because the standard options framework does not consider a hierarchical state space (in this case, the FSA). Therefore, changing the high-level FSA is like changing a variable that doesn’t exist in this baseline, which is why we did not include it in the composability experiments. We have added more of an explanation of “Flat Options” to the revision.
>
>  We can also add the RM and Flat Options baselines to the composability experiments if you think it would be helpful, if only to demonstrate their inability to be composed.
> 2) We apologize for the lack of clarity in the Methods section. We will add an outline of our approach to the beginning of the Methods section which we hope will make the connections between the subsections clearer. LTL is not used directly in the LOF algorithm, but is rather used as an aid to a user to specify the rules of the system. There are automated tools that can then be used to translate the liveness property (and safety property if applicable) into an automaton [1]. The reason we chose to use LTL to specify the task and rules (rather than, say, making an FSA by hand) is because LTL corresponds closely to natural language and has proven to be a more natural way of expressing tasks and rules for engineers than designing FSAs by hand [2]. We have added this justification to the revision. LOF also uses many concepts from the formal methods community including the ideas of propositions, liveness/safety properties, and satisfaction. But LOF does not rely on LTL – in practice, the FSA could be made by hand without using LTL.
>
>  The composability of LOF is its most unique attribute, as there are many existing algorithms that have the properties of satisfaction and optimality but not composability. LOF’s composability arises from 2facts – 1) our model is hierarchical (including the reward function), so the high level (associated with the FSA) is relatively independent from the low level (associated with the environment). The low level can be learned independently from the high level as we outline in Algorithm 1 (the low-level options are learned first (Alg 1 line 6), and only then is the metapolicy computed (Alg 1 line 13)). If the high-level FSA is modified, a new metapolicy can be calculated without needing to re-learn the low-level options. 2) We associate options with subgoals rather than with automaton states (in contrast to RM). Since these “logical options” are associated with subgoals/FSA transitions (rather than with states as in RM), they can be treated as actions over the FSA. Therefore if the FSA is changed, it is easy to recompute an optimal policy over the new FSA in 10 -50 retraining steps using Logical Value Iteration (hence, composability).
> 3) We agree that there are many papers that use FSA-based reward functions to learn tasks and have satisfaction and optimality. However, we are unaware of any of these methods being composable; therefore, in our Related Works section, these methods are discussed in the “Not Composable” paragraph.
> 4) RM is not in curve 5e because learning was so slow that its curve would have obscured the other learning curves. Figure 14 in the Appendix includes the RM learning curve. We will mention this in the caption of the figure to make it more clear, but we do discuss it in the Results section: “For the reacher domain, RM takes an order of magnitude more steps to train, so we left it out of the figure for clarity (see Appendix Fig. 14).”
> 5) Thanks for pointing out our inconsistency with FSA vs “finite state automaton”. We will fix that.
>
> [1] https://spot.lrde.epita.fr/ (for an interactive web app that converts LTL formulae to automata, see https://spot.lrde.epita.fr/app/ )
>
> [2] https://papers.ssrn.com/sol3/papers.cfm?abstract_id=3383958

---

### Decision · Program_Chairs · 2021-01-07
**Final Decision**

**Decision:**

Reject

**Comment:**

This was a borderline paper with a split recommendation from the reviewers.  The authors took great care to answer the reviewer questions in detail, and the clarity and precision of the technical exposition was strengthened.  However, substantial technical content was added to the paper during the rebuttal process, which the reviewers were not able to fully and properly assess.

Overall, this is worthwhile research, but the paper is still maturing.  The contribution was perceived as incremental in light of previous work using LTL and FSAs in RL, despite the authors extensively re-explaining the significance of the work in the rebuttal.  A resubmission is more likely to resonate with reviewers and ultimately achieve higher impact.

For completeness, it would help to also briefly acknowledge and compare to hierarchical RL work that also seeks to capture composable subtask structures, such as:

Sohn et al. "Hierarchical reinforcement learning for zero-shot generalization with subtask dependencies", NeurIPS-2018

Sohn et al. "Meta Reinforcement Learning with Autonomous Inference of Subtask Dependencies", ICLR-2020